# SAEs *Can* Improve Unlearning: Dynamic Sparse Autoencoder Guardrails for Precision Unlearning in LLMs

**Aashiq Muhamed[†], Jacopo Bonato[‡]\*, Mona Diab[†], Virginia Smith[†]**
{amuhamed,mdiab,smithv}@andrew.cmu.edu, jacopo.bonato@prometeia.com
[†]Carnegie Mellon University, [‡]Prometeia S.p.A.

## Abstract

Machine unlearning is a promising approach to improve LLM safety by removing unwanted knowledge from a trained model. However, prevailing gradient-based unlearning methods suffer from issues such as high computational costs, hyperparameter instability, poor sequential unlearning capability, vulnerability to relearning attacks, low data efficiency, and lack of interpretability. While Sparse Autoencoders are well-suited to improve these aspects by enabling targeted activation-based unlearning, prior approaches underperform gradient-based methods. This work demonstrates that, contrary to these earlier findings, SAEs can significantly improve unlearning when employed dynamically. We introduce **D**ynamic **S**AE **G**uardrails (DSG), a novel method for precision unlearning that leverages principled feature selection and a dynamic classifier. Our experiments show DSG substantially outperforms leading unlearning methods, achieving superior forget-utility trade-offs. DSG addresses key drawbacks of gradient-based approaches for unlearning—offering enhanced computational efficiency and stability, robust performance in sequential unlearning, stronger resistance to relearning attacks, better data efficiency including zero-shot settings, and more interpretable unlearning.[1]

## 1 Introduction

Machine unlearning, the process of removing specific information from trained LLMs, is a promising tool for applications in safety, privacy, and model maintenance (Liu et al., 2025). However, predominant gradient-based unlearning methods suffer from significant limitations (Barez et al., 2025). Existing methods struggle to precisely balance forgetting target data with preserving general utility (Thaker et al., 2025), incur high **computational costs** (Cha et al., 2024), exhibit **hyperparameter instability** (Bu et al., 2024), degrade quickly under **sequential unlearning** requests (Shi et al., 2024; Gao et al., 2025), are vulnerable to **relearning attacks** (Hu et al., 2025; Deeb & Roger, 2024; Łucki et al., 2024), lack **data efficiency** (Gao et al., 2024), and offer little **interpretability** (Xu et al., 2024). While interventions using Sparse Autoencoders (SAEs) (Bricken et al., 2023) offer a potential path towards more targeted, activation-based unlearning, existing SAE approaches have typically underperformed gradient-based approaches due to imprecise interventions that cause unintended side effects (Farrell et al., 2024).

This paper demonstrates that, contrary to previous findings, **SAEs *can* significantly improve unlearning** when employed dynamically. We introduce **Dynamic SAE Guardrails (DSG)**, a novel method that leverages SAEs for precise, efficient, and interpretable unlearning in LLMs. DSG integrates Fisher Information-based feature selection to identify features causally linked to the forget data, with a dynamic, input-dependent classifier that triggers targeted feature clamping only when necessary. This conditional intervention acts as a guardrail, preventing the model from accessing specific knowledge pathways for relevant

---

\*Work performed at Leonardo Labs, now at Prometeia S.p.A.

[1]Code is available at https://github.com/aashiqmuhamed/DynamicSAEGuardrails

inputs while leaving general capabilities intact. Through extensive experiments on standard benchmarks, we show that DSG not only achieves a superior balance between forgetting and utility preservation compared to leading gradient-based and static SAE methods, but also directly addresses their core limitations. Our main contributions are:

1. We introduce DSG, a new activation-based unlearning method featuring principled SAE feature selection and a dynamic classifier for precise, conditional intervention.

2. We demonstrate empirically that DSG achieves a superior balance between forgetting and utility preservation compared to leading gradient-based and SAE-based unlearning approaches on multiple benchmarks.

3. We show that DSG provides substantial benefits over gradient-based unlearning such as greater **hyperparameter stability**, improved **computational efficiency**, and **sequential unlearning capability**, enhanced **resistance against relearning attacks**, enhanced **data efficiency even in the zero-shot setting** and **interpretable unlearning**.

## 2 Background and Related Work

**Unlearning in Large Language Models.** Machine unlearning aims to modify a trained target model $\mathcal{M}(D)$ to behave as if specific data, the forget set $D_{forget}$, had never been part of its training data $D$ (Bourtoule et al., 2021; Cao & Yang, 2015). The resulting model, $\mathcal{M}_{unlearn}$, should ideally be indistinguishable from a model trained only on the retain set $D_{retain} = D \setminus D_{forget}$. As retraining LLMs from scratch is computationally prohibitive, research focuses on *approximate unlearning* (Liu et al., 2025). These methods face the core challenge of balancing knowledge removal (*forget quality*) and maintaining general capabilities (*utility preservation*) (Shi et al., 2024; Maini et al., 2024).

The dominant approach for approximate unlearning involves gradient-based optimization (Liu et al., 2024). Methods like Gradient Ascent (GA) (Jang et al., 2023), Gradient Difference (GradDiff) (Liu et al., 2022), Negative Preference Optimization (NPO) (Zhang et al., 2024), and RMU (Li et al., 2024) finetune model weights to reduce the influence of $D_{forget}$, often using regularization (e.g., KL divergence) to protect utility (Maini et al., 2024; Yao et al., 2024). However, these gradient-based techniques frequently suffer from significant limitations: high **computational cost** (requiring backward passes), instability under **hyperparameter tuning**, degraded performance under **sequential unlearning** requests (Gao et al., 2024), vulnerability to **relearning attacks** (Hu et al., 2025), poor **data efficiency**, and a lack of **interpretability** (Barez et al., 2025). These widespread challenges motivate the exploration of alternative unlearning paradigms, such as the activation-based interventions explored in this work.

**Sparse Autoencoders (SAEs).** Modern DNNs operate in a regime of superposition, where multiple features or capabilities are encoded along the same dimensions of hidden activations (Elhage et al., 2022). SAEs provide an unsupervised method for disentangling these superposed representations into interpretable features (Bricken et al., 2023; Cunningham et al., 2023). Given activations $\mathbf{h} \in \mathbb{R}^{d_{model}}$ from a specific layer or component of an LLM, an SAE decomposes and reconstructs these activations using encoder and decoder functions: $f(\mathbf{h}) := \sigma(\mathbf{W}_{enc}\mathbf{h} + \mathbf{b}_{enc})$ and $\hat{\mathbf{h}}(f) := \mathbf{W}_{dec}f + \mathbf{b}_{dec}$. In other words, SAEs express model activations as a sparse linear combination of interpretable feature vectors: $\mathbf{h} = \hat{\mathbf{h}} + \varepsilon(\mathbf{h}) = \sum_{i=1}^{d_{SAE}} f_i(\mathbf{h})\mathbf{v}_i + \mathbf{b} + \varepsilon(\mathbf{h})$ where $f_i(\mathbf{h}) \in \mathbb{R}$ are scalar feature activations, $v_i \in \mathbb{R}^{d_{model}}$ are unit vector feature directions, $b \in \mathbb{R}^{d_{model}}$ is a bias term, and $\varepsilon(\mathbf{h}) \in \mathbb{R}^{d_{model}}$ is the SAE error term. Wider SAEs continue to improve feature granularity and reduce the error term.

SAEs are trained on activations collected from the model processing pretraining data where training is conducted separately for each layer or component of interest (e.g., residual stream, attention outputs) using an objective that minimizes reconstruction loss while enforcing sparsity: $L = \|\mathbf{h} - \hat{\mathbf{h}}(f(\mathbf{h}))\|_2^2 + \lambda\|f(\mathbf{h})\|_0$. Here $\lambda$ is a sparsity penalty coefficient encouraging most feature activations to be zero for any given input. In this work, we use JumpReLU SAEs (Rajamanoharan et al., 2024), which enforce sparsity using a shifted Heaviside step function. The interpretability of SAE features stems from their sparse activation

pattern—because features are only active for a small fraction of inputs, they must capture meaningful patterns to be useful for reconstruction. The cost of training SAEs is amortized across multiple downstream applications such as identifying and removing spurious correlations in models (Marks et al., 2024) and steering behavior (O'Brien et al., 2024).

Farrell et al. (2024) developed an early approach using SAEs for unlearning by identifying features that were frequently active on $D_{forget}$ and applying a *static* intervention—clamping these identified features when active at a token irrespective of overall context. However, this produced substantial side-effects on general model utility and ultimately underperformed gradient-based methods like RMU. In this work, we show that SAE unlearning *can* be effective via a context-dependent intervention strategy rather than simple static clamping.

*Further background details are provided in Appendix A.*

## 3 Dynamic SAE Guardrails (DSG)

DSG (Figure 1, Algorithm 1) is a targeted unlearning method for LLMs that leverages the interpretability of SAEs and combines: (1) a causal framing that motivates feature selection, (2) a theoretically justified feature importance scoring based on Fisher Information (FI), (3) a dynamic, input-dependent classification rule based on a statistically optimal threshold, and (4) a targeted clamping intervention to remove the influence of selected features.

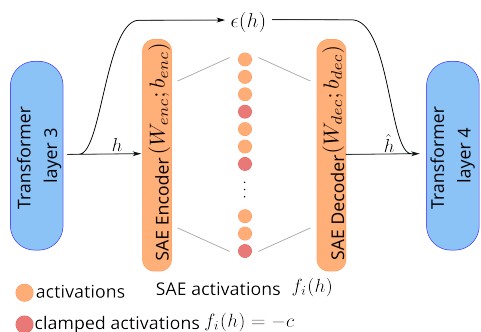

**Figure 1:** An illustration of DSG

### 3.1 Causal Framework and Problem Setup

We frame unlearning through a causal perspective where $\mathcal{D}_{\text{forget}}$ and $\mathcal{D}_{\text{retain}}$ influence model representations $\mathbf{E}$ and outputs $\mathbf{Y}$ through multiple pathways (Shen et al., 2024). These include representation-mediated pathways ($\mathcal{D} \to \mathbf{E} \to \mathbf{Y}$), potential direct influence ($\mathcal{D} \to \mathbf{Y}$), and intertwined knowledge ($\mathcal{D}_{\text{forget}} \leftrightarrow \mathcal{D}_{\text{retain}}$) where conceptual overlap exists between the datasets. SAE features $f_j$ derived from $\mathbf{E}$ serve as interpretable mediators (Pearl, 2009) of information flow through these causal pathways. From this perspective, unlearning involves blocking pathways from $\mathcal{D}_{\text{forget}}$ to $\mathbf{Y}$ while preserving the pathways from $\mathcal{D}_{\text{retain}}$ to $\mathbf{Y}$. DSG implements this causal intervention $\text{do}(f_j = -c)$ on forget features identified by analyzing SAE activation patterns across both datasets.

### 3.2 Feature Selection: Identifying Causal Mediators via Fisher Information

To identify which SAE features, $F_{\text{forget}}$ mediate the causal influence of $\mathcal{D}_{\text{forget}}$, we establish two key theoretical connections: first between FI and feature activation, and second between FI and causal influence. We then describe our percentile based feature selection.

**Theorem 1** (Fisher Information Approximation). *For an SAE with small reconstruction error and input $\mathbf{h}$, the expected squared gradient of reconstruction loss with respect to feature $j$'s decoder weights $\boldsymbol{\theta}_{j,\cdot}$ is proportional to the second moment of that feature's activation: $\mathbb{E}_{\mathbf{h}}[\|\nabla_{\boldsymbol{\theta}_{i,\cdot}} \ell(\mathbf{h})\|^2] \approx \epsilon^2 \mathbb{E}_{\mathbf{h}}[f_j(\mathbf{h})^2]$ where w.h.p, reconstruction errors are bounded by $\epsilon^2$.*

The proof of Theorem 1 is provided in Appendix B. This establishes that squared feature activations are proportional to the Fisher Information of the corresponding decoder weights.

**Theorem 2** (Fisher Information as a Proxy for Causal Feature Importance). *Under standard assumptions, Fisher Information associated with SAE features provides an approximation of their causal influence as mediators between specific training data and model outputs. For any SAE feature $f_j$, the expected squared activation $\mathbb{E}_D[f_j(\mathbf{h})^2]$ on dataset $D$ is proportional to the causal influence of that feature as a mediator of information from $D$ to model outputs.*

Proof of Theorem 2 is in Appendix C. Under these assumptions, a feature with large expected squared activation on $\mathcal{D}_{\text{forget}}$ contributes significantly to the model's FI *with respect to that data*. This suggests that intervening on that feature (e.g., clamping its activation) would substantially affect the model's output distribution when processing inputs similar to those in $\mathcal{D}_{\text{forget}}$. Squared activation serves as a computationally tractable proxy for causal influence.

**Importance Scores.** DSG obtains token-level SAE activations from each sequence in both $\mathcal{D}_{\text{forget}}$ and $\mathcal{D}_{\text{retain}}$, squares them, and aggregates the results into matrices $\mathbf{A}_{\text{forget}} \in \mathbb{R}^{n_F \times d_{\text{SAE}}}$ and $\mathbf{A}_{\text{retain}} \in \mathbb{R}^{n_R \times d_{\text{SAE}}}$ ($n_F$ and $n_R$ are the total numbers of tokens in the respective datasets). For each token $t$ in sequence $x$, we have the activation $f_j(\mathbf{h}_{x,t})$ of feature $j$ on the hidden state $\mathbf{h}_{x,t}$. Each entry of the activation matrices is thus $\mathbf{A}_{\text{forget}}[i,j] \approx \left[f_j(\mathbf{h}_{x,t})\right]^2$ for a token $t$ in sequence $x \in \mathcal{D}_{\text{forget}}$ (and similarly for $\mathbf{A}_{\text{retain}}$). From these, we compute the average squared activation per feature as $\texttt{forget\_score}(j) = 1/n_F \sum_{x \in \mathcal{D}_{\text{forget}}} \sum_{t=1}^{|x|} \left[f_j(\mathbf{h}_{x,t})\right]^2$ and $\texttt{retain\_score}(j) = 1/n_R \sum_{x \in \mathcal{D}_{\text{retain}}} \sum_{t=1}^{|x|} \left[f_j(\mathbf{h}_{x,t})\right]^2$, and define the relative importance by $\texttt{imp\_ratio}(j) = \frac{\texttt{forget\_score}(j)}{\max\{\texttt{retain\_score}(j), \varepsilon\}}$, with $\varepsilon > 0$ to avoid division by zero. By Theorem 2 this ratio represents the relative causal influence of feature $j$.

---

**Algorithm 1 Dynamic SAE Guardrails (DSG)**

---

**Require:** LLM with SAE features $\{f_j\}$; datasets $\mathcal{D}_{\text{forget}}, \mathcal{D}_{\text{retain}}$; clamp strength $c$; percentiles $(p_{\text{ratio}}, p_{\text{dyn}})$; feature count $n_{\text{feats}}$
    **Feature Selection:**
        Compute feature importance scores and threshold $\tau_{\text{ratio}}$ from percentiles
        Identify $F_{\text{forget}} = \{j : \texttt{imp\_ratio}(j) \geq \tau_{\text{ratio}}\}$
        Sort $F_{\text{forget}}$ by descending $\texttt{forget\_score}(j)$ and select top $n_{\text{feats}}$ features to form $S_{n_{\text{feats}}}$
    **Dynamic Threshold Calibration:**
        Compute $\rho(x) = \frac{1}{|x|} \sum_t \mathbf{1}[\exists j \in S_{n_{\text{feats}}} : f_j(\mathbf{h}_t) > 0]$ for each $x \in \mathcal{D}_{\text{retain}}$
        Set threshold $\tau = \text{Percentile}(\{\rho(x)\}_{x \in \mathcal{D}_{\text{retain}}}, p_{\text{dyn}})$
    **Inference-Time Intervention:**
        For input sequence $x$, compute $\rho(x)$ and classify as forget-relevant if $\rho(x) > \tau$
        If forget-relevant: For each token $t$ and feature $j \in S_{n_{\text{feats}}}$, set $f_j'(\mathbf{h}_t) = -c$
        Otherwise: Preserve all feature activations

---

**Percentile-Based Feature Selection.** To select features most causally relevant to $\mathcal{D}_{\text{forget}}$, we employ a percentile-based approach using $p_{\text{ratio}}$ to compute $\tau_{\text{ratio}}$ as $\text{Percentile}(\{\texttt{imp\_ratio}(j)\}_{j=1}^{d_{\text{SAE}}}, p_{\text{ratio}})$. $\text{Percentile}(S, p)$ returns the value $v$ such that $p\%$ of elements in set $S$ are less than or equal to $v$. For example, with $p_{\text{ratio}} = 95$, $\tau_{\text{ratio}}$ is set so 95% of features have $\texttt{imp\_ratio}(j) \leq \tau_{\text{ratio}}$. We define the set of forget-mediating features as $F_{\text{forget}} = \{j : \texttt{imp\_ratio}(j) \geq \tau_{\text{ratio}}\}$. To filter out noisy features, we sort features in $F_{\text{forget}}$ by descending $\texttt{forget\_score}(j)$ and select the top $n_{\text{feats}}$ to form the final intervention set $S_{n_{\text{feats}}}$.

## 3.3 Dynamic Sequence-Level Classification and Intervention

DSG employs a dynamic input-dependent classification mechanism to minimize unintended side-effects on content unrelated to the forget knowledge.

**Definition 1** (Forget-Set Activated Token). *A token $x_t$ is considered* forget-set activated *if at least one feature $j \in S_{n_{\text{feats}}}$ has a positive activation: $f_j(\mathbf{h}_t) > 0$.*

For an input sequence $x = (x_1, \ldots, x_T)$ of length $T$, we define the statistic $\rho(x) = \frac{1}{T} \sum_{t=1}^{T} \mathbf{1}[\exists j \in S_{n_{\text{feats}}} : f_j(\mathbf{h}_t) > 0]$, representing the percentage of forget-set activated tokens. A high $\rho(x)$ indicates that query $x$ strongly relies on features we've identified as causally linked to the forget knowledge.

**Threshold Selection and Classification.** We select a threshold $\tau \in [0, 1]$ based on the distribution of $\rho(x)$ on $\mathcal{D}_{\text{retain}}$ using $\tau = \text{Percentile}(\{\rho(x)\}_{x \in \mathcal{D}_{\text{retain}}}, p_{\text{dyn}})$ which controls the trade-off between unlearning effectiveness and performance preservation.

Empirically, we find $\rho(x)$ is stochastically larger on $\mathcal{D}_{\text{forget}}$ than on $\mathcal{D}_{\text{retain}}$, as seen in Figure 2, which shows the distribution for both forget-domain queries (WMDP-Bio) and general knowledge queries (MMLU). $\tau$ is chosen to control the retain set's false-positive rate and separates forget-set queries effectively, achieving high recall on $\mathcal{D}_{\text{forget}}$. On this example, DSG successfully transfers from retain set (WikiText) and forget set to the test query set. We define classifier $C(x) = \mathbf{1}[\rho(x) > \tau]$, labeling inputs as either *forget-relevant* or *retain-relevant*. The statistical optimality of this thresholding approach follows from Neyman-Pearson lemma:

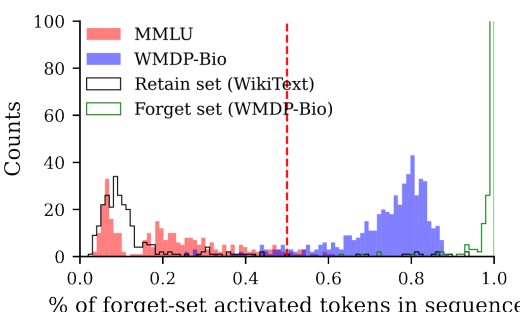

**Figure 2:** Distribution of $\rho(x)$ for unlearning on WMDP-Bio. Threshold at 95th percentile (dashed red line) separates MMLU from WMDP.

**Theorem 3** (Neyman-Pearson Optimality). *If $\rho(X)$ is stochastically larger under $\mathcal{D}_{\text{forget}}$ than under $\mathcal{D}_{\text{retain}}$, then among all classifiers with a false-positive rate at most $\alpha$, the threshold test $C^*(x) = \mathbf{1}[\rho(x) > \tau^*]$, where $\Pr_{X \sim \mathcal{D}_{\text{retain}}}[\rho(X) > \tau^*] = \alpha$, maximizes the true-positive rate.*
The proof appears in Appendix D and states that under the stochastic dominance assumption, thresholding $\rho(x)$ is the optimal classification approach for a given false-positive rate.

**Conditional Clamping.** Our intervention is conditional on the classifier $C(x)$. When $C(x) = 1$ (forget-relevant), for each token $x_t$ and feature $j \in S_{n_{\text{feats}}}$, we set $f'_j(\mathbf{h}_t) = -c$, where $-c$ is a large negative constant we call *clamp strength*. This implements a targeted $do(f_j(\mathbf{h}_t) = -c)$ operation, selectively severing the causal pathway *only when the input query is deemed forget-relevant*. When $C(x) = 0$ (retain-relevant), we leave all features unchanged: $f'_j(\mathbf{h}_t) = f_j(\mathbf{h}_t)$. This preserves the model's original behavior for queries unrelated to the targeted knowledge, minimizing side-effects and maintaining performance on $\mathcal{D}_{\text{retain}}$.

The dynamic clamping in DSG contrasts with static clamping methods (Farrell et al., 2024), which intervene based only on feature activation without sequence-level classification, and risk inadequate coverage on $\mathcal{D}_{\text{forget}}$ or excessive side-effects on $\mathcal{D}_{\text{retain}}$. DSG avoids this suboptimal trade-off—we formally prove (Theorem 4, Appendix E) that for any static approach, DSG achieves equal or greater coverage on $\mathcal{D}_{\text{forget}}$ with equivalent side-effects on $\mathcal{D}_{\text{retain}}$, **providing a superior unlearning-utility trade-off**.

## 4 Experiments and Results

### 4.1 Unlearning on WMDP

We evaluate DSG on the WMDP dataset (Li et al., 2024), which benchmarks hazardous knowledge unlearning across multiple domains. We focus on WMDP-Bio (biosecurity) and WMDP-Cyber (cybersecurity). For each domain, our unlearning setup uses domain-specific $\mathcal{D}_{\text{forget}}$—PubMed papers containing bio-weapon related content for WMDP-Bio and GitHub repositories for WMDP-Cyber—and WikiText (Merity et al., 2016) as $\mathcal{D}_{\text{retain}}$. We evaluate unlearning effectiveness using the WMDP multiple-choice question test sets, which were not exposed to models during the unlearning process.

Following SAEBench (Karvonen et al., 2025), we evaluate unlearning only on questions the target model correctly answers across all 24 permutations of the 4 multiple-choice options. This yields 522/1273 questions for WMDP-Bio and 275/1987 questions for WMDP-Cyber. For evaluating model utility, we similarly filter MMLU questions that the model answers correctly across all permutations. This yields 305 questions from history, computer science, geography, and human aging for WMDP-Bio. For WMDP-Cyber, we use 371 MMLU questions, replacing computer science with biology. Table 1 reports the configuration that minimizes WMDP accuracy while maintaining at least 99% of the target model MMLU accuracy along with with MT-Bench scores that measure general fluency.

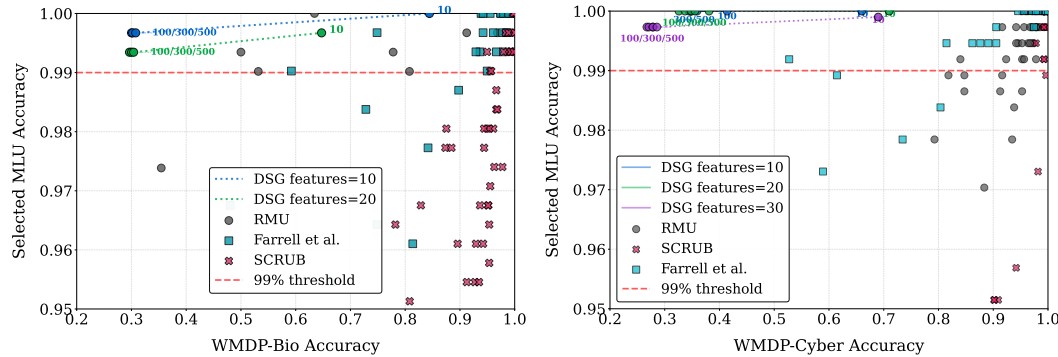

**Figure 3:** Unlearning performance on WMDP-Bio (left) and WMDP-Cyber (right). Higher MMLU accuracy and lower WMDP accuracy is better. Clamp strengths ($c$) used for DSG points are shown as annotations. DSG Pareto-dominates the top four baseline methods (RMU, SCRUB, Farrell et al., SSD).

| Method | WMDP Bio ($\downarrow$) | MMLU ($\uparrow$) | | | | | MT ($\uparrow$) |
|---|---|---|---|---|---|---|---|
| | | HS Hist | C. CS | HS Geo | H. Aging | All | |
| Target $\mathcal{M}$ | 100.00 | 100.00 | 100.00 | 100.00 | 100.00 | 100.00 | 7.36 |
| GA | 99.44 | 98.18 | 100.00 | 100.00 | 100.00 | 99.35 | 7.44 |
| NPO | 97.95 | 99.99 | 88.88 | 100.00 | 98.82 | 99.35 | 7.29 |
| SSD | 99.44 | 100.00 | 100.00 | 100.00 | 98.82 | 99.68 | 7.24 |
| SCRUB | 94.97 | 99.09 | 100.00 | 100.00 | 98.82 | 99.35 | 6.09 |
| Farrell et al. | 59.22 | 100.00 | 100.00 | 100.00 | 96.47 | 99.03 | 7.33 |
| RMU | 50.00 | 99.08 | 100.00 | 100.00 | 98.81 | 99.47 | 7.21 |
| **DSG (Ours)** | **29.64** | 100.00 | 100.00 | 100.00 | 97.62 | 99.34 | **7.78** |

**Table 1:** Unlearning performance on WMDP-Bio. All represents the average MMLU score. MT-Bench scores show 0.16 variance across 5 runs. DSG shows superior unlearning effectiveness compared to baselines while maintaining high MMLU performance.

**Experimental Setup.** We implement DSG using `gemma-2-2b-it` model with `gemma-scope-2b-pt-res` SAE (width 16k) (Lieberum et al., 2024) applied to layer 3 at $\ell_0$ 142. We use $P_{dyn} = 95$ for both domains, and $P_{ratio} = 95$ for WMDP-Bio and $P_{ratio} = 90$ for WMDP-Cyber. We compare DSG against several baselines across a broad hyperparameter sweep: GA (Jang et al., 2023), NPO (Zhang et al., 2024), SSD (Foster et al., 2024), SCRUB (Kurmanji et al., 2023), Farrell et al. (2024) and RMU (Li et al., 2024). Complete hyperparameter details are provided in Appendix G.

**Results.** As shown in Table 1, DSG significantly outperforms all baselines on the WMDP-Bio unlearning task, reducing accuracy to 29.64% compared to the next best method RMU at 50.00%. It maintains high MMLU performance (99.34% average) and achieves the highest MT-Bench score (7.78), showing superior preservation of general model capabilities. The results on WMDP-Cyber (Appendix G.2) reinforce these findings. Figure 3 provides a more comprehensive view of the unlearning-utility trade-off landscape, plotting all configurations with MMLU accuracy above 95%. DSG Pareto-dominates all baseline methods: for any level of utility preservation (MMLU accuracy), DSG achieves more effective unlearning.

This superior performance is coupled with significant practical advantages over gradient-based methods in terms of **computational efficiency and hyperparameter stability**. Gradient-based approaches often exhibit **hyperparameter instability**, where slight tuning changes can drastically alter outcomes, risking poor unlearning or utility collapse. Furthermore, they require computationally costly backward passes through the LLM for optimization. In contrast, DSG shows greater **hyperparameter stability** (Figure 3) and efficiency. It requires only forward passes: one to gather feature statistics initially, and then lightweight intervention during inference, completely avoiding expensive gradient calculations. This combination of efficiency and stability makes DSG particularly advantageous for large models and frequent unlearning where gradient computations impose substantial overhead.

## 4.2 Unlearning on Muse

We evaluate DSG on MUSE (Shi et al., 2024) comprising two corpora: NEWS and BOOKS, and focusing on six dimensions: verbatim memorization, knowledge memorization, privacy leakage, utility preservation, forget set scalability, and sequential unlearning. MUSE provides a challenging evaluation setting where forget and retain sets share substantial domain overlap: in NEWS, both sets are drawn from the same distribution of BBC articles, while in BOOKS, the forget set (Harry Potter books) and retain set (Harry Potter FanWiki) contain highly related content.

| | C1. No Verbatim Mem. VerbMem on $\mathcal{D}_{\text{forget}}$ (↓) | | C2. No Knowledge Mem. KnowMem on $\mathcal{D}_{\text{forget}}$ (↓) | | C3. No Privacy Leak. PrivLeak ($\in [-5\%, 5\%]$) | | C4. Utiltiy Preserv. KnowMem on $\mathcal{D}_{\text{retain}}$ (↑) | |
|---|---|---|---|---|---|---|---|---|
| | | | | **NEWS** | | | | |
| Target $\mathcal{M}$ | 21.15 | | 29.51 | | −88.16 | | 26.78 | |
| GA | 0.62 | ↓ 97.1% | 0.00 | ↓ 100.0% | -8.16 | under-unlearn | 0.09 | ↓ 99.7% |
| GradDiff | 2.81 | ↓ 86.7% | 0.71 | ↓ 97.6% | 93.10 | over-unlearn | 7.76 | ↓ 71.0% |
| NPO | 20.98 | ↓ 0.8% | 25.14 | ↓ 14.8% | -53.42 | under-unlearn | 29.02 | ↑ 8.4% |
| SimNPO | 21.14 | ↓ 0.0% | 27.70 | ↓ 6.1% | -89.84 | under-unlearn | 30.59 | ↑ 14.2% |
| RMU | 9.60 | ↓ 54.6% | 26.63 | ↓ 9.8% | 75.02 | over-unlearn | 26.41 | ↓ 1.4% |
| **DSG (Ours)** | 11.80 | ↓ 44.2% | 0.44 | ↓ 98.5% | 12.08 | over-unlearn | 25.65 | ↓ 4.2% |
| | | | | **BOOKS** | | | | |
| Target $\mathcal{M}$ | 15.80 | | 33.90 | | −98.80 | | 35.28 | |
| GA | 2.61 | ↓ 83.5% | 0.17 | ↓ 99.5% | -1.58 | acceptable | 0.57 | ↓ 98.4% |
| GradDiff | 9.49 | ↓ 39.9% | 21.57 | ↓ 36.4% | -10.30 | under-unlearn | 23.66 | ↓ 32.9% |
| NPO | 14.41 | ↓ 8.8% | 28.21 | ↓ 16.8% | -97.24 | under-unlearn | 37.19 | ↑ 5.4% |
| SimNPO | 14.55 | ↓ 7.9% | 34.36 | ↑ 1.4% | -96.40 | under-unlearn | 36.62 | ↑ 3.8% |
| RMU | 14.89 | ↓ 5.8% | 32.59 | ↓ 3.9% | -97.58 | under-unlearn | 37.13 | ↑ 5.2% |
| **DSG (Ours)** | 8.73 | ↓ 44.7% | 1.79 | ↓ 94.7% | -23.18 | under-unlearn | 37.10 | ↑ 5.2% |

**Table 2:** Unlearning performance on MUSE. We highlight in green if the method satisfies the criterion and red otherwise. For privacy leakage, large positive values suggest over-unlearning, while large negative values suggest under-unlearning. DSG shows strong performance across all metrics, achieving substantial reductions in verbatim and knowledge memorization while maintaining high utility.

**Experimental Setup.** We create separate target models for NEWS and BOOKS by finetuning gemma-2-2b-it on each corpus for 5 epochs using learning rate $10^{-5}$ and batch size 32. For each target model, we implement DSG using gemma-scope-2b-pt-res SAE (width 16k) applied to layer 3. For both domains, we use clamp strength 500, $p_{ratio} = 95$ and $n_{\text{feats}} = 20$. We use $p_{dyn} = 90$ for NEWS and $p_{dyn} = 95$ for BOOKS, with the lower threshold for NEWS enabling more effective verbatim memorization removal. For both scalability and sequential unlearning, we use the best NEWS hyperparameters.

We compare DSG against: GA, GradDiff (Liu et al., 2022), NPO, SimNPO (Fan et al., 2024), and RMU. Following MUSE, we train for 10 epochs using AdamW with learning rate $10^{-5}$ and batch size 32, selecting the last epoch checkpoint before utility falls below 90% of the target model accuracy. Complete hyperparameters can be found in Appendix H.

**Unlearning.** Table 2 shows that DSG outperforms existing baselnes. It is effective at verbatim memorization removal (C1) with 44.2% reduction on NEWS and 44.7% on BOOKS. On knowledge memorization (C2), DSG achieves near-complete removal with 98.5% reduction on NEWS and 94.7% reduction on BOOKS, outperforming most baselines. On privacy leakage (C3), while not within the ideal range, DSG performs better than the majority of baselines. For utility preservation (C4), DSG maintains 95.8% of target model performance on NEWS and achieves a 5.2% improvement on BOOKS compared to the target model.

**Scalability.** Figure 4(a) shows **DSG is stable and robust when scaling to larger forget sets**. We evaluate performance across forget sets ranging from 0.8M to 3.3M tokens, and DSG maintains its position in the ideal region (high retain set knowledge, low forget set knowledge) even as the forget set size increases. In contrast, gradient-based methods exhibit substantial degradation, with increasingly poor tradeoffs between retaining general knowledge and forgetting targeted information.

**Sequential Unlearning.** Figure 4(b) illustrates **DSG's effectiveness across sequential unlearning requests** on four disjoint NEWS folds. We implement two approaches: $DSG_{all}$, which cumulatively updates feature importance scores based on each new forget data request; and $DSG_{union}$, which takes the union of features selected independently at each step and uses this combined set to calculate $\rho(x)$ and threshold $\tau$ on $D_R$. Both approaches perform similarly well, consistently maintaining

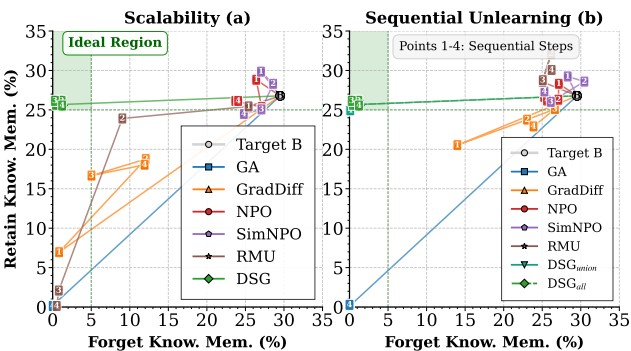

**Figure 4:** (a) Scalability: Performance across increasing forget set sizes. (b) Sequential Unlearning: Performance across sequential unlearning requests

DSG in the ideal region where other methods rapidly degrade with each additional unlearning operation. Gradient-based methods suffer from catastrophic forgetting during sequential unlearning, where each update pushes the model further from its original performance distribution. (Details in subsection H.2.)

## 4.3 Resistance to Relearning Attacks

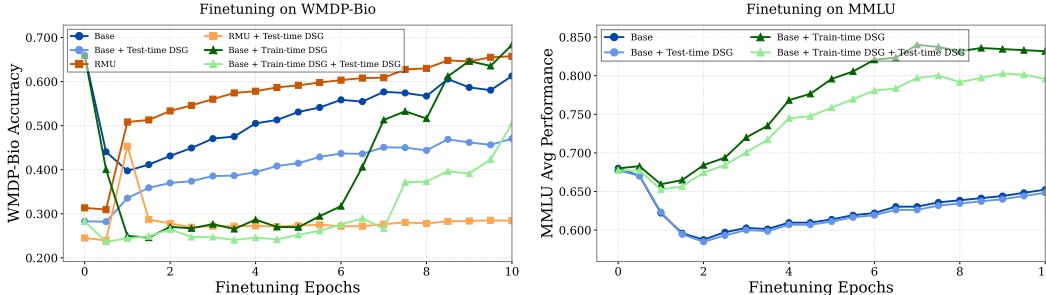

**Figure 5:** Relearning attack resistance across finetuning epochs. (a) DSG demonstrates superior resistance to relearning compared to RMU. (b) Test-time DSG preserves MMLU utility better than Train-time DSG while still providing significant protection.

We evaluate DSG's resistance to relearning attacks in API-based threat models where adversaries have query access but cannot directly manipulate model weights. This resistance derives from the Superficial Alignment Hypothesis (Zhou et al., 2023), which posits that a model's activation geometry stabilizes during pretraining and changes minimally during finetuning. Figure 9a confirms this empirically, showing high cosine similarity between pre-finetuning and post-finetuning activation vectors, and activation magnitudes clustered around 1.0. By operating on these stable activation patterns rather than weights, DSG creates a more persistent defense. While obfuscation-based attacks have been proposed against activation-based interventions (Bailey et al., 2024), they are less effective in API-based black-box settings where attackers lack direct access to gradients and model representations.

**Methodology.** We evaluate two DSG defenses against relearning: (1) Test-time DSG, which applies intervention only at inference time after model finetuning, and (2) Train-time DSG, which integrates DSG during finetuning with frozen SAE parameters to filter gradients. We test six configurations with `google/gemma-2-2b-it` as base model: Base, Base+Test-time DSG, Base+Train-time DSG, Base+Train-time DSG+Test-time DSG, RMU (base model with RMU unlearning applied), and RMU+Test-time DSG. The relearning attack consists of finetuning each configuration on the WMDP-Bio test set for 10 epochs at learning rate 1e-5.

**Results and Analysis.** Figure 5(a) demonstrates clear differences in vulnerability to relearning attacks. Weight-based methods show high susceptibility, with RMU rapidly increasing

in WMDP-Bio accuracy when finetuned, eventually exceeding the base model's finetuned performance. The base model itself shows an initial performance decrease before increasing, as the high learning rate temporarily undoes instruction tuning before relearning occurs.

**Test-time DSG** provides substantial protection, with RMU+Test-time DSG maintaining near-random performance (25%) throughout training. However, Base+Test-time DSG shows gradual vulnerability to relearning, with performance slowly increasing over finetuning epochs. This gradual protection erosion reveals a limitation of test-time intervention alone.

**Train-time DSG** offers a distinct protective mechanism. Models finetuned with DSG active show immediate reduction to random-level performance that persists through approximately six epochs before gradually recovering. This delayed recovery pattern suggests DSG forces the model to develop entirely new processing circuits rather than simply reactivating suppressed knowledge. Figure 9b supports this interpretation, showing significantly higher training loss on WMDP-Bio compared to MMLU when finetuning with DSG active.

**Combining both approaches** (Train-time DSG+Test-time DSG) extends resistance through epoch 7, demonstrating how these complementary mechanisms can be layered for enhanced protection. However, these approaches involve utility trade-offs. Figure 5(b) shows that while Base Finetuned and Base Finetuned+Test-time DSG maintain comparable MMLU performance, Train-time DSG exhibits moderate utility decline at higher epoch counts.

DSG's **superior resistance to relearning attacks** stems from its activation-based intervention that leverages the stability of activation geometry during finetuning.

## 4.4 Data Efficiency and Zero-shot Interpretable Unlearning

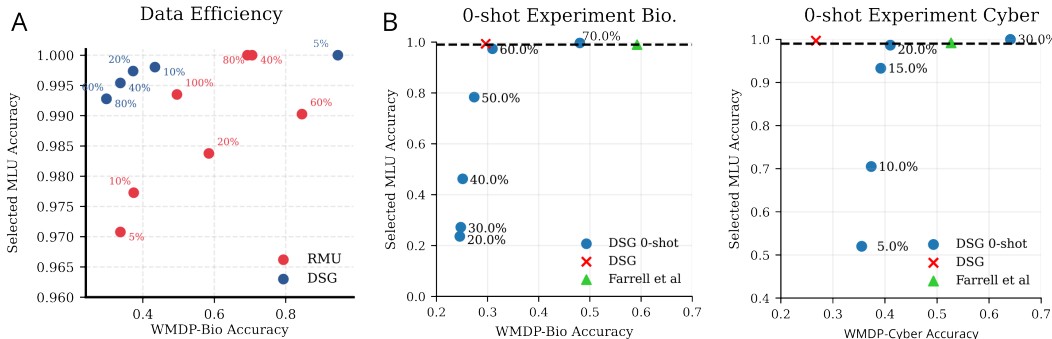

**Figure 6:** Data efficiency analysis of DSG. (A) Performance across varying training data sizes compared to RMU. (B) Zero-shot performance on WMDP-Bio (left) and WMDP-Cyber (right) using 20 features selected via Neuropedia API with different $\tau$ thresholds (shown next to each data point).

We evaluate how DSG performs with limited forget and retain data on WMDP-Bio. Figure 6A shows DSG maintaining consistent performance when trained on 20-80% of the original retain and forget datasets, preserving MMLU accuracy while keeping WMDP accuracy below 40%. Only when dataset size falls below 20% does effectiveness noticeably decline, with WMDP accuracy rising above 40%. In contrast, RMU shows inconsistent results across different dataset sizes, indicating that gradient-based methods may be more **unstable to hyperparameter changes** when data is limited.

For **zero-shot evaluation**, we implement DSG without any domain-specific forget or retain data (Figure 6B), instead **leveraging the interpretability of SAEs**. We use Neuropedia (Lin, 2023) feature explanations to identify the forget set features by querying for concepts *Biology* and *Cybersecurity*, selecting the top 20 relevant features (details in Appendix M). Both tasks use the gemma-scope-2b-pt-res SAE (width 16k) at layer 3 ($\ell_0$ 59). Since retain data is unavailable for dynamic threshold calibration, we sweep over static $\tau$ values, finding optimal settings ($\tau = 60\%$ for WMDP-Bio, $\tau = 20\%$ for WMDP-Cyber). Even with features selected purely based on their semantic descriptions and without dataset-specific tuning beyond $\tau$, these zero-shot DSG configurations outperform RMU and Farrell et al. (2024), demonstrating the potential for effective unlearning guided directly by feature interpretability.

## 4.5 Ablations

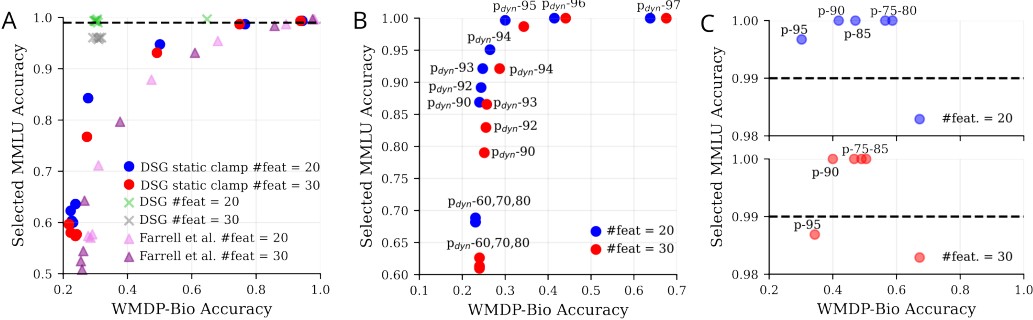

**Figure 7:** DSG Ablation studies (A) Static vs. dynamic clamping comparison with varying clamp strengths [10-500] for 20 and 30 features. (B) Effect of dynamic threshold percentile ($p_{\text{dyn}}$) on performance (C) Impact of importance ratio threshold ($p_{\text{ratio}}$, range 75-95) for 20 and 30 features.

We evaluate each component of DSG by conducting ablation experiments on WMDP-Bio.

**Dynamic Classification:** Figure 7A compares DSG with dynamic classification against DSG with static clamping from Farrell et al. (2024) While static clamping effectively removes forget-set information at large clamp values ($c > 100$), it simultaneously reduces MMLU accuracy because it treats all inputs identically regardless of their forget-relevance. In contrast, our dynamic classifier only applies interventions when necessary based the statistical distribution of forget-feature activations. This conditional approach maintains higher MMLU accuracy ($> 99\%$) while achieving comparable or better WMDP-Bio reduction.

**Percentile-Based Feature Selection:** DSG with static clamping leverages Fischer Information for feature selection instead of feature sparsity in Farrell et al. (2024). As shown in Figure 7A, across equivalent clamping strengths, this selection approach achieves 8% lower WMDP-Bio accuracy on average while maintaining comparable MMLU performance, indicating more precise identification of forget-relevant features.

**Dynamic Threshold $p_{\text{dyn}}$:** Figure 7B shows the effect of $p_{\text{dyn}}$ on the forget-retain trade-off. Higher percentiles ($> 95$) preserve more MMLU accuracy but allow more WMDP content to pass through undetected, while lower percentiles ($< 90$) apply intervention more aggressively but with increased side effects on general knowledge. The optimal range 90-95 balances these considerations, removing targeted knowledge while minimizing side effects.

**Importance Ratio Threshold $p_{\text{ratio}}$:** As shown in Figure 7C, varying $p_{\text{ratio}}$ from 75-95 provides fine-grained control over feature selection. Higher values (95) select features with stronger forget-retain differentiation, yielding more targeted intervention, while lower values expand the feature set but may increase overlap with general knowledge features. Additionally we observed that the dynamic classifier can compensate for a lower $p_{\text{ratio}}$ maintaining effective forget-set filtering even when feature selection is less discriminative.

Additional ablations in Appendix K show that DSG is remarkably robust to clamp strength variations, and performs optimally with moderate feature counts. These findings highlight DSG's **practical hyperparameter stability.** Effective performance is maintained within reliable ranges for thresholds $p_{\text{dyn}}/p_{\text{ratio}}$ (90-95), feature counts (10-20), alongside notable robustness to clamp strength (100-500). Additionally, these hyperparameters transfer across datasets, simplifying deployment compared to gradient-based methods.

## 5 Conclusion and Future Work

In this work, we introduced DSG, demonstrating that SAEs with dynamic classification enable precise, activation-based unlearning that substantially outperforms gradient-based methods across multiple benchmarks. While our evaluation focuses on Gemma-2-2B where high-quality SAEs are available, extending DSG to larger models and diverse architectural families remains an important avenue for future work, as does investigating how performance scales with SAE width and training configurations.

## Acknowledgements

This work was supported in part by the National Science Foundation grants IIS2145670 and CCF2107024, and funding from Amazon, Apple, Google, Intel, Meta, the CyLab Security and Privacy Institute, and Leonardo Labs. Any opinions, findings and conclusions, or recommendations expressed in this material are those of the author(s) and do not necessarily reflect the views of any of these funding agencies.

## Ethics Statement

This work introduces Dynamic SAE Guardrails (DSG), a method for targeted unlearning in large language models (LLMs). While designed to promote responsible AI by enabling the removal of unwanted knowledge, several ethical considerations arise:

- **Potential for misuse:** While our focus is on removing hazardous or unwanted knowledge, the same technology could potentially be used to censor information or suppress viewpoints, leading to undesirable social consequences if deployed without careful oversight. The zero-shot capabilities, while advantageous for data-scarce scenarios, could be misused if the user-provided keywords are biased or used to target specific groups/content unfairly.

- **Over-reliance on interpretability:** Although SAEs offer improved interpretability compared to black-box models, feature interpretations are not always definitive or fully reliable. Misinterpreting feature roles or over-relying on imperfect interpretations could lead to unintended consequences, including the removal of valuable knowledge or the failure to remove harmful content. The quality of feature interpretation depends on the quality and representativeness of the data used to train and interpret the SAE.

- **Limitations of unlearning:** As with all approximate unlearning methods, DSG does not guarantee complete removal of targeted knowledge. As we show, it reduces the likelihood of the model generating outputs related to the forget set, but subtle traces or indirect influences might persist. It is essential to acknowledge these limitations and avoid presenting DSG as a perfect solution for knowledge removal.

- **Dual-use concerns:** The techniques developed in this work for improving model control and safety could also be adapted by malicious actors to develop more sophisticated attacks or to create models that resist safety interventions. We recognize this inherent dual-use nature and emphasize the need for responsible development and sharing of research findings.

- **Computational Cost of SAE Training:** The training of SAEs can be computationally demanding, raising environmental concerns. However there are several open-source SAEs, amortizing their cost, and the the inference-time efficiency of DSG offers some mitigation compared to gradient-based unlearning approaches.

We believe the benefits of precise, controllable unlearning for enhancing AI safety outweigh these risks, provided the technology is developed and deployed responsibly. We encourage future work to address these limitations and explore more robust evaluation methods for unlearning.

## Reproducibility Statement

To facilitate reproducibility, we have provided detailed descriptions of our experimental setups, including all relevant datasets, models, and hyperparameters. Specifics for each set of experiments can be found as follows:

- **WMDP Unlearning:** Complete hyperparameter settings for DSG and all baseline methods (GA, NPO, SSD, SCRUB, Farrell et al., RMU) are detailed in Appendix G.1. Model and SAE details are provided in Section 4.1.

- **MUSE Unlearning:** Hyperparameters for DSG and baseline methods (GA, GradDiff, NPO, SimNPO, RMU) are in Appendix H.1, with model details in Section 3.

- **Relearning Attacks (Section 4.3):** Model and hyperparameter configurations for the relearning experiments, including both train-time and test-time DSG interventions, are given in Appendix I.5, along with the details in the main text.

- **Data Efficiency and Zero-shot Experiments:** Model, SAE, and hyperparameter details for the data efficiency analysis and zero-shot evaluations are given in Appendix J.

- **Ablations:** All details related to the ablation studies, and chosen hyperparameters are in Appendix K.

We have described the feature selection process, dynamic classification rule, and intervention mechanism in sufficient detail to allow for reimplementation (Algorithm 1 and Section 3). We have also released the code and relevant scripts necessary to reproduce our results.

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

# A  Additional Background and Related Work Details

This appendix provides further details on concepts mentioned in the Background and Related Work (Section 2).

## A.1  Formal Goal of Unlearning

As introduced in the main text, machine unlearning aims to transform a model $\mathcal{M}(D)$, initially trained on a dataset $D = D_{retain} \cup D_{forget}$, into an unlearned model $\mathcal{M}_{unlearn}$. The theoretical ideal of *exact unlearning* requires that $\mathcal{M}_{unlearn}$ be computationally indistinguishable from a model $\mathcal{M}(D_{retain})$ that was trained exclusively on the retain set $D_{retain}$ from the beginning (Bourtoule et al., 2021; Cao & Yang, 2015). Due to the computational cost of retraining large language models from scratch, achieving exact unlearning is generally impractical. Therefore, the field primarily focuses on developing *approximate unlearning* methods. These methods aim to satisfy specific criteria related to effectively removing the influence of $D_{forget}$ while preserving the model's performance on $D_{retain}$, without incurring the cost of full retraining (Liu et al., 2025).

## A.2  Unlearning Evaluation Metrics and Benchmarks

The evaluation of approximate unlearning methods typically involves measuring two primary aspects: Forget Quality and Utility Preservation. **Forget Quality** quantifies the successful removal of information pertaining to the forget set $D_{forget}$. Common metrics include measuring the *Forget Set Performance Degradation*, which involves observing reduced accuracy or increased loss on tasks specifically related to the content of $D_{forget}$ (Shi et al., 2024; Maini et al., 2024). Another aspect is assessing *Memorization Metrics*, which gauge the model's reduced ability to recall specific sequences or knowledge points verbatim from $D_{forget}$ (Shi et al., 2024). Furthermore, *Privacy Leakage Metrics* evaluate the decreased success rate of Membership Inference Attacks (MIAs) that try to infer whether a given data point was part of the original $D_{forget}$, often quantified using the Area Under the Curve (AUC) of the MIA classifier (Shokri et al., 2017; Shi et al., 2024).

Conversely, **Utility Preservation** assesses how well the unlearned model retains its general knowledge and capabilities on tasks unrelated to $D_{forget}$. This is commonly measured by evaluating *Retain Set Performance Preservation*, which checks for maintained accuracy on standard academic or commonsense reasoning benchmarks such as MMLU (Hendrycks et al., 2020). Additionally, *General Language Modeling Performance* is often assessed by ensuring minimal increase in perplexity or loss when the model processes large, general-purpose text corpora like OpenWebText (Gokaslan et al., 2019) or WikiText (Merity et al., 2016). Finally, *Fluency and Coherence* of the model's generated text are important, often evaluated through automated metrics, human judgment, or interaction with benchmark chatbots like MT-Bench (Zheng et al., 2023). Standardized benchmarks like MUSE (Shi et al., 2024), TOFU (Maini et al., 2024), WMDP (Li et al., 2024), and SAEBench (Karvonen et al., 2025) provide datasets, tasks, and evaluation protocols designed to measure performance across these diverse criteria.

## A.3  Gradient-Based Unlearning Methods

Gradient-based unlearning techniques directly modify the weights $\theta$ of the original model $\mathcal{M}(D)$ using optimization algorithms, typically variants of gradient descent or ascent.

**Gradient Ascent (GA)** represents a basic approach where the optimization objective is to maximize the loss function (e.g., negative log-likelihood) on the forget set $D_{forget}$, thereby pushing the model parameters away from configurations that accurately represent this data (Jang et al., 2023). This method, however, often suffers from catastrophic forgetting of useful knowledge if not carefully regularized.

**Gradient Difference (GradDiff or NegGrad)** attempts to balance forgetting and retention by computing gradients for both minimizing loss on $D_{retain}$ and maximizing loss on $D_{forget}$,

then applying an update based on a combination (often a subtraction) of these gradients (Liu et al., 2022).

**Negative Preference Optimization (NPO)** leverages insights from preference-based fine-tuning methods like DPO (Rafailov et al., 2023), reformulating unlearning as learning to disprefer outputs related to $D_{forget}$ relative to some reference, which could be outputs from the original model or data from $D_{retain}$ (Zhang et al., 2024). Simplified variants like SimNPO aim to reduce the computational overhead (Fan et al., 2024).

**Representation Misdirection Unlearning (RMU)** operates by injecting noise or applying targeted shifts to the internal activations of the model at specific layers, but only when processing inputs related to $D_{forget}$, while simultaneously using a regularization term to keep activations on $D_{retain}$ close to those of the original model (Li et al., 2024).

**Selective Synaptic Dampening (SSD)** aims for more targeted weight modification by estimating the importance of individual parameters for both $D_{forget}$ and $D_{retain}$ (using approximations based on Fisher information) and then selectively reducing the magnitude of parameters found to be more critical for $D_{forget}$ than for $D_{retain}$ (Foster et al., 2024).

**SCRUB** employs a student-teacher knowledge distillation framework; it trains a copy of the original model (the student) to diverge from the original frozen model (the teacher) on $D_{forget}$ inputs (typically by maximizing KL divergence) while simultaneously encouraging the student to mimic the teacher on $D_{retain}$ inputs (by minimizing KL divergence) (Kurmanji et al., 2023).

Finally, many gradient-based methods incorporate explicit **Regularization Techniques** to counteract the tendency towards catastrophic forgetting. Common regularizers include minimizing the KL divergence between the probability distributions of the unlearned and original models when evaluated on $D_{retain}$ (Maini et al., 2024), or directly including a term in the loss function that minimizes the model's prediction error on $D_{retain}$ (Yao et al., 2024).

### A.4   Prior SAE Unlearning Work

The work by Farrell et al. (2024) is an early exploration into using Sparse Autoencoders (SAEs) for machine unlearning. We describe their methodology here.

First, they computed the activation sparsity for each feature in the SAE dictionary, calculated separately over the forget dataset ($D_{forget}$) and the retain dataset ($D_{retain}$). Sparsity here refers to the fraction of input tokens for which a given feature has a non-zero activation.

Second, to mitigate potential damage to the model's general capabilities, they filtered out any features whose activation sparsity on the retain set $D_{retain}$ exceeded a predetermined threshold (e.g., a feature active on more than 1% of retain tokens might be excluded).

Third, from the pool of features that passed the retain-sparsity filter, they selected the top-$N$ features that exhibited the highest activation sparsity when measured on the forget set $D_{forget}$. The assumption was that features frequently active on forget data are likely responsible for encoding the knowledge to be removed.

Fourth, they implemented a *static* intervention mechanism during inference: whenever any of the top-$N$ selected features $f_j$ produced a positive activation ($f_j(\mathbf{h}_t) > 0$) for any token $t$, its activation was clamped to a fixed negative value (e.g., -c). This clamping was applied universally, regardless of the overall context of the input sequence.

This combination of sparsity-based feature selection and static clamping ultimately proved limiting, leading to significant side effects on utility and performance inferior to contemporary gradient-based methods like RMU on benchmarks such as WMDP-Bio. Recognizing these limitations directly motivated our work (DSG), where we instead develop and apply principled feature selection and dynamic, context-aware interventions.

## A.5 Relearning Attacks

Approximate unlearning methods, especially those modifying model weights, face another significant challenge: **relearning attacks** (Deeb & Roger, 2024). In these attacks, an adversary finetunes the unlearned model $\mathcal{M}_{unlearn}$ to recover the supposedly forgotten information. Such recovery is sometimes possible even using only data tangentially related to the original forget set $D_{forget}$ (Hu et al., 2025). The success of relearning attacks suggests that gradient-based weight modifications may primarily suppress access to knowledge rather than truly erasing it from the parameter space; subsequent finetuning can often reverse these weight adjustments, particularly if it reinforces the target concepts.

The feasibility of relearning attacks strongly depends on the threat model. In an **API-based (black-box) setting**, where adversaries only have query access, mounting effective relearning attacks is more difficult, particularly if the provider restricts extensive finetuning or monitors queries. Activation-based intervention methods like DSG, which modify internal states rather than weights to control outputs for relevant inputs, may offer greater robustness in this black-box scenario compared to weight modification techniques.

Although sophisticated **obfuscation attacks** targeting activation-based defenses exist (Bailey et al., 2024), they typically require white-box access (e.g., gradients, internal states). Such access is unavailable in a pure API setting, limiting their threat against deployed systems focused on output safety via activation manipulation. DSG's potential resilience against relearning could stem from the relative stability of activation geometry during standard finetuning, a phenomenon related to the **Superficial Alignment Hypothesis** (Zhou et al., 2023). If DSG reliably identifies features encoding the target knowledge based on these stable patterns and consistently applies interventions, the unlearning effect may prove more durable against finetuning-based relearning attacks compared to methods reliant on weight configurations.

## B  Fisher Information Approximation Proof

**Theorem 1** (Approximate Fisher Information from SAE Features). *Let a sparse autoencoder (SAE) with reconstruction $\hat{\mathbf{r}}(x) = \mathbf{z}(x)\mathbf{W}_{\text{dec}}$ be applied to data $x \sim \mathcal{D}$, where $\mathbf{z}(x) \in \mathbb{R}^F$ represents latent activations and $\mathbf{W}_{\text{dec}} \in \mathbb{R}^{F \times D}$ the decoder weights. Define the reconstruction loss as:*

$$\ell(x) = \frac{1}{2}\|\hat{\mathbf{r}}(x) - \mathbf{r}(x)\|^2$$

*If the SAE is well-trained such that reconstruction error is small with high probability, then for each row $\boldsymbol{\theta}_{i,\cdot} \in \mathbb{R}^D$ of $\mathbf{W}_{\text{dec}}$ (representing feature i), the expected squared gradient is approximately proportional to the second moment of the feature activation.*

*Proof.* We establish this result through careful analysis of the gradient structure in sparse autoencoders.

**Computing the Gradient of Decoder Weights.**  By definition of the reconstruction loss:

$$\ell(x) = \frac{1}{2}\|\hat{\mathbf{r}}(x) - \mathbf{r}(x)\|^2$$
$$= \frac{1}{2}\|\mathbf{z}(x)\mathbf{W}_{\text{dec}} - \mathbf{r}(x)\|^2$$

For row $i$ of $\mathbf{W}_{\text{dec}}$, denoted $\boldsymbol{\theta}_{i,\cdot} \in \mathbb{R}^D$, we compute the gradient:

$$\nabla_{\boldsymbol{\theta}_{i,\cdot}}\ell(x) = \nabla_{\boldsymbol{\theta}_{i,\cdot}}\left[\frac{1}{2}\|\mathbf{z}(x)\mathbf{W}_{\text{dec}} - \mathbf{r}(x)\|^2\right]$$

By the chain rule:

$$\nabla_{\boldsymbol{\theta}_{i,\cdot}}\ell(x) = (\mathbf{z}(x)\mathbf{W}_{\text{dec}} - \mathbf{r}(x)) \cdot \nabla_{\boldsymbol{\theta}_{i,\cdot}}(\mathbf{z}(x)\mathbf{W}_{\text{dec}})$$

Since $\mathbf{z}(x)\mathbf{W}_{\mathrm{dec}}$ is linear in $\boldsymbol{\theta}_{i,\cdot}$ with coefficient $z_i(x)$, we have:

$$\nabla_{\boldsymbol{\theta}_{i,\cdot}}(\mathbf{z}(x)\mathbf{W}_{\mathrm{dec}}) = z_i(x) \cdot \mathbf{I}_D$$

where $\mathbf{I}_D$ is the $D$-dimensional identity matrix. Therefore:

$$\nabla_{\boldsymbol{\theta}_{i,\cdot}}\ell(x) = z_i(x)(\hat{\mathbf{r}}(x) - \mathbf{r}(x))$$

**Computing the Squared Gradient Norm.**  Taking the squared norm of this gradient:

$$\|\nabla_{\boldsymbol{\theta}_{i,\cdot}}\ell(x)\|^2 = \|z_i(x)(\hat{\mathbf{r}}(x) - \mathbf{r}(x))\|^2$$
$$= z_i(x)^2 \|\hat{\mathbf{r}}(x) - \mathbf{r}(x)\|^2$$

Taking the expectation over the data distribution:

$$\mathbb{E}_{x\sim\mathcal{D}}[\|\nabla_{\boldsymbol{\theta}_{i,\cdot}}\ell(x)\|^2] = \mathbb{E}_{x\sim\mathcal{D}}[z_i(x)^2 \|\hat{\mathbf{r}}(x) - \mathbf{r}(x)\|^2]$$

**Analyzing the Small Error Regime.**  When the SAE is well-trained, we can characterize its performance with a high-probability bound on reconstruction error. Specifically, assume there exist constants $\epsilon > 0$ and $\delta > 0$ such that:

$$\mathbb{P}\left(\|\hat{\mathbf{r}}(x) - \mathbf{r}(x)\|^2 < \epsilon^2\right) > 1 - \delta$$

where $\epsilon \ll \|\hat{\mathbf{r}}(x)\|$ and $\delta$ is small. In other words, the squared reconstruction error is bounded by $\epsilon^2$ with probability at least $1 - \delta$.

Under this high-probability bound, we can decompose the expectation:

$$\mathbb{E}[z_i(x)^2\|\hat{\mathbf{r}}(x) - \mathbf{r}(x)\|^2] \le \mathbb{E}[z_i(x)^2 \cdot \epsilon^2 \mid \|\hat{\mathbf{r}}(x) - \mathbf{r}(x)\|^2 < \epsilon^2] \cdot (1 - \delta) + C\delta$$
$$\le \epsilon^2 \mathbb{E}[z_i(x)^2] + C\delta$$

where $C$ is a bound on the expectation in the low-probability case. For small $\delta$ and finite $C$, the second term becomes negligible, leaving:

$$\mathbb{E}[z_i(x)^2\|\hat{\mathbf{r}}(x) - \mathbf{r}(x)\|^2] \approx \epsilon^2 \mathbb{E}[z_i(x)^2]$$

**Connection to Fisher Information.**  The Fisher Information Matrix for parameter $\boldsymbol{\theta}_{i,\cdot}$ is defined as:

$$\mathcal{I}(\boldsymbol{\theta}_{i,\cdot}) = \mathbb{E}_{x\sim\mathcal{D}}[\nabla_{\boldsymbol{\theta}_{i,\cdot}}\ell(x)\nabla_{\boldsymbol{\theta}_{i,\cdot}}\ell(x)^\top]$$

The trace of this matrix, which measures the overall sensitivity of the loss to changes in $\boldsymbol{\theta}_{i,\cdot}$, is precisely:

$$\mathrm{Tr}(\mathcal{I}(\boldsymbol{\theta}_{i,\cdot})) = \mathbb{E}_{x\sim\mathcal{D}}[\|\nabla_{\boldsymbol{\theta}_{i,\cdot}}\ell(x)\|^2]$$
$$\approx \epsilon^2 \mathbb{E}[z_i(x)^2]$$

**Interpretation.**  The above analysis shows that $(f_j(x))^2 = z_j(x)^2$ serves as a natural importance measure for feature $j$. Features with larger average squared activations contribute more significantly to reconstruction gradients and thus have higher Fisher Information content. This justifies our approach of using squared activations to identify features most strongly associated with specific knowledge domains. □

## C  Connecting Fisher Information to Causal Influence

In this section, we establish how the Fisher Information associated with Sparse Autoencoder (SAE) features connects to their causal influence as mediators of information flow in language models. Drawing inspiration from causal geometry (Chvykov & Hoel, 2020), we provide a proof for why expected squared activation serves as a measure of feature importance.

**Theorem 2** (Fisher Information as a Proxy for Causal Feature Importance). *Under assumptions of (i) near-deterministic mappings in the language model, (ii) well-defined causal effects under feature interventions, (iii) small SAE reconstruction error, and (iv) approximate feature independence, the Fisher Information associated with SAE features provides a principled approximation of their causal influence. Specifically, for any feature $f_j$, the expected squared feature activation $\mathbb{E}_{\mathcal{D}}[f_j(\mathbf{h})^2]$ for hidden state $\mathbf{h}$ on dataset $\mathcal{D}$ is proportional to the causal influence of that feature as a mediator of information from $\mathcal{D}$ to model outputs.*

*Proof.* We build upon the result in Appendix B, which showed that the expected squared activation $\mathbb{E}[f_j(\mathbf{h})^2]$ is proportional to the trace of the Fisher Information Matrix for the corresponding decoder weights.

**Causal Model Setup.** Consider a language model (LM) that produces hidden states $\mathbf{h}(x) \in \mathbb{R}^d$. A Sparse Autoencoder (SAE) encodes $\mathbf{h}(x)$ into feature activations $\mathbf{z} = f(\mathbf{h}) \in \mathbb{R}^{d_{\text{SAE}}}$, i.e. each feature is $f_j(\mathbf{h}(x))$. Let $\mathcal{D}_{\text{forget}}$ and $\mathcal{D}_{\text{retain}}$ be two subsets of the training data. We model the causal structure as:

$$\text{Data} \longrightarrow \mathbf{h}(x) \longrightarrow \mathbf{z} = f(\mathbf{h}) \longrightarrow \mathbf{Y} \text{ (model outputs)}$$

Here, $\mathbf{Y} \in \mathbb{R}^{d_Y}$ represents the model's output vector (e.g., logits or embeddings).

**Assumptions.**

1. **Near-deterministic mapping.** Conditioned on $\mathbf{h}$, the model output $\mathbf{Y}$ is almost deterministic (small Gaussian noise). Formally, $p(\mathbf{Y} \mid \mathbf{h}) \approx \mathcal{N}(\boldsymbol{\mu}(\mathbf{h}), \sigma^2 \mathbf{I})$ with small $\sigma^2$.

2. **Well-defined feature interventions.** We can perform $do(f_j = \alpha)$, meaning forcibly setting feature $j$ to $\alpha$ and thus severing its normal dependence on $\mathbf{h}$.

3. **Small SAE reconstruction error.** Writing $\hat{\mathbf{h}}(\mathbf{z}) \approx \mathbf{W}\mathbf{z}$, we assume $\|\mathbf{h} - \hat{\mathbf{h}}(\mathbf{z})\|$ is small with high probability.

4. **Approximate feature independence.** Features $f_j(\mathbf{h})$ are sufficiently sparse or decorrelated that cross-terms can be neglected.

**Defining Causal Influence.** We quantify the causal influence of feature $f_j$ by how much the model's output distribution $p(\mathbf{Y})$ changes when we intervene to set $f_j$ to its normal value $f_j(\mathbf{h})$ vs. forcing it to zero:

$$\text{Influence}(f_j) = \mathbb{E}_{\mathbf{h} \sim \mathcal{D}}\Big[ D_{\text{KL}}\Big( p\big(\mathbf{Y} \mid do(f_j = f_j(\mathbf{h}))\big) \,\|\, p\big(\mathbf{Y} \mid do(f_j = 0)\big) \Big) \Big]$$

A large KL means toggling $f_j$ from 0 to its actual value drastically shifts $p(\mathbf{Y})$, so $f_j$ is a strong mediator for $\mathcal{D}$.

**Expanding KL Divergence.** Let $\mathbf{g}_j : \mathbb{R} \to \mathbb{R}^{d_Y}$ describe how feature $f_j$ shifts the model's outputs. Since we forcibly set $f_j$ (an intervention), we ignore any prior correlations with $\mathbf{h}$, and under near-determinism the output distribution is approximated by:

$$p\big(\mathbf{Y} \mid do(f_j = \alpha)\big) = \mathcal{N}\big(\mathbf{g}_j(\alpha), \, \sigma^2 \mathbf{I}\big)$$

For two different interventions $do(f_j = \alpha)$ and $do(f_j = \beta)$, we can now derive the KL divergence between the resulting output distributions. Using the standard formula for KL divergence between multivariate Gaussians with the same covariance matrix:

$$D_{\text{KL}}(\mathcal{N}(\boldsymbol{\mu}_1, \boldsymbol{\Sigma}) \| \mathcal{N}(\boldsymbol{\mu}_2, \boldsymbol{\Sigma})) = \frac{1}{2}(\boldsymbol{\mu}_1 - \boldsymbol{\mu}_2)^T \boldsymbol{\Sigma}^{-1}(\boldsymbol{\mu}_1 - \boldsymbol{\mu}_2)$$

Therefore:

$$
\begin{aligned}
D_{\mathrm{KL}}\Big(p(\mathbf{Y}\mid do(f_j = \alpha)) \,\big\|\, p(\mathbf{Y}\mid do(f_j = \beta))\Big) &= D_{\mathrm{KL}}(\mathcal{N}(\mathbf{g}_j(\alpha), \sigma^2 \mathbf{I}) \| \mathcal{N}(\mathbf{g}_j(\beta), \sigma^2 \mathbf{I})) \\
&= \frac{1}{2}(\mathbf{g}_j(\alpha) - \mathbf{g}_j(\beta))^T (\sigma^2 \mathbf{I})^{-1} (\mathbf{g}_j(\alpha) - \mathbf{g}_j(\beta)) \\
&= \frac{1}{2\sigma^2}(\mathbf{g}_j(\alpha) - \mathbf{g}_j(\beta))^T (\mathbf{g}_j(\alpha) - \mathbf{g}_j(\beta)) \\
&= \frac{1}{2\sigma^2} \left\| \mathbf{g}_j(\alpha) - \mathbf{g}_j(\beta) \right\|^2
\end{aligned}
$$

**First-Order Taylor Expansion.** To make this expression more tractable, we use a first-order Taylor expansion of $\mathbf{g}_j(\alpha)$ around $\beta = 0$:

$$
\begin{aligned}
\mathbf{g}_j(\alpha) &= \mathbf{g}_j(0) + \frac{d\mathbf{g}_j}{d\alpha}\Big|_{\alpha=0} \cdot \alpha + o(\alpha) \\
&\approx \mathbf{g}_j(0) + \left(\nabla \mathbf{g}_j(0)\right) \alpha
\end{aligned}
$$

When $\alpha$ is sufficiently small, the higher-order terms $o(\alpha)$ become negligible. Substituting this back into our KL divergence expression for the special case where $\beta = 0$:

$$
\begin{aligned}
D_{\mathrm{KL}}\left(p(\mathbf{Y}|do(f_j = \alpha)) \| p(\mathbf{Y}|do(f_j = 0))\right) &= \frac{1}{2\sigma^2} \|\mathbf{g}_j(\alpha) - \mathbf{g}_j(0)\|^2 \\
&\approx \frac{1}{2\sigma^2} \|\mathbf{g}_j(0) + \nabla \mathbf{g}_j(0) \cdot \alpha - \mathbf{g}_j(0)\|^2 \\
&= \frac{1}{2\sigma^2} \|\nabla \mathbf{g}_j(0) \cdot \alpha\|^2 \\
&= \frac{1}{2\sigma^2} \alpha^2 \|\nabla \mathbf{g}_j(0)\|^2
\end{aligned}
$$

This shows that the KL divergence (our measure of distribution change) grows quadratically with the intervention magnitude $\alpha$, with a proportionality constant determined by the gradient norm $\|\nabla \mathbf{g}_j(0)\|^2$.

**Expected Causal Influence.** Now we can compute the expected causal influence by substituting $\alpha = f_j(\mathbf{h})$ and taking the expectation over $\mathbf{h} \sim \mathcal{D}$:

$$
\begin{aligned}
\mathrm{Influence}(f_j) &= \mathbb{E}_{\mathbf{h} \sim \mathcal{D}} \left[ D_{\mathrm{KL}}\left(p(\mathbf{Y}|do(f_j = f_j(\mathbf{h}))) \| p(\mathbf{Y}|do(f_j = 0))\right) \right] \\
&\approx \mathbb{E}_{\mathbf{h} \sim \mathcal{D}} \left[ \frac{1}{2\sigma^2} f_j(\mathbf{h})^2 \|\nabla \mathbf{g}_j(0)\|^2 \right] \\
&= \frac{\|\nabla \mathbf{g}_j(0)\|^2}{2\sigma^2} \mathbb{E}_{\mathbf{h} \sim \mathcal{D}}\left[ f_j(\mathbf{h})^2 \right]
\end{aligned}
$$

Thus, the expected causal influence of feature $j$ as a mediator of information from dataset $\mathcal{D}$ is directly proportional to the expected squared activation $\mathbb{E}_{\mathcal{D}}\left[f_j(\mathbf{h})^2\right]$, with a proportionality constant $\frac{\|\nabla \mathbf{g}_j(0)\|^2}{2\sigma^2}$ that depends on the sensitivity of the model's outputs to changes in feature $j$.

**Connection to Fisher Information.** The Fisher Information for the SAE's decoder weights $\boldsymbol{\theta}_{j,\cdot}$ satisfies $\mathcal{I}(\boldsymbol{\theta}_{j,\cdot}) \propto \mathbb{E}_{\mathbf{h}}\left[f_j(\mathbf{h})^2\right]$ since the gradient w.r.t. $\boldsymbol{\theta}_{j,\cdot}$ includes $f_j(\mathbf{h})$ as a leading factor. Therefore, $\mathbb{E}[f_j(\mathbf{h})^2]$ tracks both the Fisher Information and the intervention-based notion of causal influence we derived above, establishing a direct link: Causal Influence$(f_j) \propto$ Fisher Information$(\boldsymbol{\theta}_{j,\cdot}) \propto \mathbb{E}[f_j(\mathbf{h})^2]$. In other words, features most *important* in a Fisher Information sense are precisely those with greatest causal influence on model outputs.

**Implications for Feature Selection.** By identifying features with high squared activations on $\mathcal{D}_{\text{forget}}$ but low activations on $\mathcal{D}_{\text{retain}}$, we can target mediators that specifically carry forget set knowledge. Clamping these features to zero during inference selectively reduces the model's capacity to propagate information from $\mathcal{D}_{\text{forget}}$ while preserving performance on $\mathcal{D}_{\text{retain}}$.

**Comparisons Across Datasets.** For $\mathcal{D}_{\text{forget}}$ vs. $\mathcal{D}_{\text{retain}}$, we earlier defined:

$$\texttt{forget\_score}(j) = \mathbb{E}_{\mathcal{D}_{\text{forget}}}\big[f_j(\mathbf{h})^2\big]$$

$$\texttt{retain\_score}(j) = \mathbb{E}_{\mathcal{D}_{\text{retain}}}\big[f_j(\mathbf{h})^2\big]$$

The ratio of causal influence of feature $j$ for $\mathcal{D}_{\text{forget}}$ versus $\mathcal{D}_{\text{retain}}$ is:

$$
\frac{\mathbb{E}_{\mathcal{D}_{\text{forget}}}\big[\text{Influence}(f_j)\big]}{\mathbb{E}_{\mathcal{D}_{\text{retain}}}\big[\text{Influence}(f_j)\big]} = \frac{\frac{\|\nabla \mathbf{g}_j(0)\|^2}{2\sigma^2} \cdot \mathbb{E}_{\mathcal{D}_{\text{forget}}}\big[f_j(\mathbf{h})^2\big]}{\frac{\|\nabla \mathbf{g}_j(0)\|^2}{2\sigma^2} \cdot \mathbb{E}_{\mathcal{D}_{\text{retain}}}\big[f_j(\mathbf{h})^2\big]}
$$

$$
= \frac{\mathbb{E}_{\mathcal{D}_{\text{forget}}}\big[f_j(\mathbf{h})^2\big]}{\mathbb{E}_{\mathcal{D}_{\text{retain}}}\big[f_j(\mathbf{h})^2\big]}
$$

$$
= \frac{\texttt{forget\_score}(j)}{\texttt{retain\_score}(j)}
$$

Thus, $\texttt{forget\_score}(j)/\texttt{retain\_score}(j)$ precisely captures how much more $f_j$ mediates the forget dataset relative to the retain dataset. This is the importance ratio we defined in Section 3, which directly quantifies the relative causal influence of feature $j$ across datasets. $\qquad\square$

## D  Proof of Neyman-Pearson Optimality

**Theorem 3** (Neyman-Pearson Optimality). *Let $\mathcal{D}_{forget}$ and $\mathcal{D}_{retain}$ be the distributions of sequences from the forget and retain sets, respectively. If $\rho(X)$ is stochastically larger under $\mathcal{D}_{forget}$ than under $\mathcal{D}_{retain}$ (i.e., $\Pr_{X\sim\mathcal{D}_{forget}}[\rho(X) \geq t] \geq \Pr_{X\sim\mathcal{D}_{retain}}[\rho(X) \geq t]$ for all $t$), then among all classifiers with a false-positive rate at most $\alpha$, the threshold test $C^*(x) = \mathbf{1}[\rho(x) > \tau^*]$, where $\Pr_{X\sim\mathcal{D}_{retain}}[\rho(X) > \tau^*] = \alpha$, maximizes the true-positive rate.*

*Proof.* We adapt the classical Neyman-Pearson Lemma to our unlearning context. Our goal is to find the optimal decision rule for classifying inputs as either forget-relevant or retain-relevant.

Consider the class of all decision rules $a : \mathcal{X} \to \{\text{clamp}, \text{no-clamp}\}$ with false-positive rate at most $\alpha$. That is, all rules $a$ such that:

$$\Pr_{X\sim\mathcal{D}_{\text{retain}}}[a(X) = \text{clamp}] \leq \alpha$$

For each decision rule $a$, define its acceptance region $A = \{x \in \mathcal{X} : a(x) = \text{clamp}\}$. The constraint on false-positive rate translates to $\Pr_{X\sim\mathcal{D}_{\text{retain}}}[A] \leq \alpha$.

Now, define the threshold-based decision rule $a^*$ as:

$$a^*(x) = \mathbf{1}[\rho(x) > \tau^*]$$

where $\tau^*$ is chosen such that $\Pr_{X\sim\mathcal{D}_{\text{retain}}}[\rho(X) > \tau^*] = \alpha$. The acceptance region for this rule is $A^* = \{x : \rho(x) > \tau^*\}$.

We need to prove that $a^*$ maximizes the true-positive rate among all rules with false-positive rate at most $\alpha$. In other words, for any rule $a$ with $\Pr_{X\sim\mathcal{D}_{\text{retain}}}[a(X) = \text{clamp}] \leq \alpha$, we must show:

$$\Pr_{X\sim\mathcal{D}_{\text{forget}}}[a(X) = \text{clamp}] \leq \Pr_{X\sim\mathcal{D}_{\text{forget}}}[a^*(X) = \text{clamp}]$$

We use the stochastic dominance property: for any threshold $t$, $\Pr_{X \sim \mathcal{D}_{\text{forget}}}[\rho(X) \geq t] \geq \Pr_{X \sim \mathcal{D}_{\text{retain}}}[\rho(X) \geq t]$. This means that regions of higher $\rho$ values are relatively more likely under $\mathcal{D}_{\text{forget}}$ than under $\mathcal{D}_{\text{retain}}$.

Consider any decision rule $a$ with acceptance region $A$ where $\Pr_{X \sim \mathcal{D}_{\text{retain}}}[A] \leq \alpha$. Due to the stochastic dominance property, we can always construct a threshold-based region $\tilde{A} = \{x : \rho(x) > \tilde{\tau}\}$ such that: 1. $\Pr_{X \sim \mathcal{D}_{\text{retain}}}[\tilde{A}] = \Pr_{X \sim \mathcal{D}_{\text{retain}}}[A]$ (same false-positive rate) 2. $\Pr_{X \sim \mathcal{D}_{\text{forget}}}[\tilde{A}] \geq \Pr_{X \sim \mathcal{D}_{\text{forget}}}[A]$ (equal or higher true-positive rate)

This is because exchanging points from low-$\rho$ regions in $A$ with points from high-$\rho$ regions outside $A$ (while maintaining the same false-positive rate) will always increase the true-positive rate due to stochastic dominance.

If $\Pr_{X \sim \mathcal{D}_{\text{retain}}}[A] < \alpha$, we can further expand $\tilde{A}$ to $A^*$ by lowering the threshold from $\tilde{\tau}$ to $\tau^*$, which only increases the true-positive rate further.

Therefore, for any decision rule $a$ with false-positive rate at most $\alpha$:

$$\Pr_{X \sim \mathcal{D}_{\text{forget}}}[a(X) = \text{clamp}] = \Pr_{X \sim \mathcal{D}_{\text{forget}}}[A] \leq \Pr_{X \sim \mathcal{D}_{\text{forget}}}[A^*] = \Pr_{X \sim \mathcal{D}_{\text{forget}}}[a^*(X) = \text{clamp}]$$

This establishes that the threshold test $a^*(x) = \mathbf{1}[\rho(x) > \tau^*]$ maximizes the true-positive rate among all tests with false-positive rate at most $\alpha$. $\square$

**Practical Implications:** This theorem establishes the statistical optimality of our thresholding approach for making the binary decision of whether to apply an intervention. In particular, it shows that our dynamic classification rule maximizes coverage on forget-set queries while maintaining a controlled false-positive rate on retain-set queries.

## E    Proof of Dynamic Clamping Dominance

**Theorem 4** (Dominance of Dynamic Clamping). *Let $S_{n_{\text{feats}}}$ be a fixed subset of features identified as forget-relevant. Consider the static approach $a_{\text{static}}(x)$ that clamps features in $S_{n_{\text{feats}}}$ whenever they activate, and the dynamic approach $a_{\text{dynamic}}(x)$ that first classifies input $x$ using $C(x) = \mathbf{1}[\rho(x) > \tau]$ and only then applies clamping. Under the stochastic dominance assumption from Theorem 3, there exists a threshold $\tau^*$ such that $a_{\text{dynamic}}$ achieves equal or greater coverage on $\mathcal{D}_{\text{forget}}$ than $a_{\text{static}}$ while maintaining equal side-effects on $\mathcal{D}_{\text{retain}}$, making dynamic clamping strictly dominant in the coverage-side effect trade-off.*

*Proof.* We begin by formalizing the metrics used to evaluate both approaches and precisely defining their operation.

**Preliminaries and Definitions.**    Let $\mathcal{X}$ be the space of possible input sequences. For a sequence $x = (x_1, \ldots, x_T)$ and its corresponding hidden states $\mathbf{h}_t$, we define:

- A token $t$ is triggered by $S_{n_{\text{feats}}}$ if $\exists j \in S_{n_{\text{feats}}}$ such that $f_j(\mathbf{h}_t) > 0$

- The fraction of triggered tokens in a sequence: $\rho(x) = \frac{1}{T} \sum_{t=1}^{T} \mathbf{1}[\exists j \in S_{n_{\text{feats}}} : f_j(\mathbf{h}_t) > 0]$

We consider two distributions: $\mathcal{D}_{\text{forget}}$: The distribution of forget-relevant queries, and $\mathcal{D}_{\text{retain}}$: The distribution of retain-relevant queries.

**The Two Approaches.**    For both approaches, we define a *clamp set* $B_{\text{method}} \subseteq \mathcal{X}$ as the set of inputs where the method applies some clamping.

1. **Static Approach** ($a_{\text{static}}$): Clamps features in $S_{n_{\text{feats}}}$ whenever they activate on any token. here the Clamp set is $B_{\text{stat}} = \{x : \exists t, j \in S_{n_{\text{feats}}} \text{ such that } f_j(\mathbf{h}_t) > 0\}$.

2. **Dynamic Approach** ($a_{\text{dynamic}}$): Computes $\rho(x)$ and applies a threshold test $\rho(x) > \tau$. Only clamps if the sequence passes this test. The Clamp set for threshold $\tau$: $B_{\text{dyn}}(\tau) = \{x : \rho(x) > \tau\}$.

**Performance Metrics.**   We define:

- **Coverage**: The probability that clamping occurs on forget-set queries

$$\text{Coverage(method)} = \Pr_{x \sim \mathcal{D}_{\text{forget}}}[x \in B_{\text{method}}]$$

- **Side-effect**: The probability that clamping occurs on retain-set queries

$$\text{SideEffect(method)} = \Pr_{x \sim \mathcal{D}_{\text{retain}}}[x \in B_{\text{method}}]$$

**Step 1: Find the side-effect of the static approach.**   The static approach clamps whenever any token has an activating feature in $S_{n_{\text{feats}}}$. Therefore:

$$\text{SideEffect}(a_{\text{static}}) = \Pr_{x \sim \mathcal{D}_{\text{retain}}}[x \in B_{\text{stat}}] = \alpha$$

**Step 2: Find a threshold $\tau^*$ that yields the same side-effect for the dynamic approach.** By our assumption that $\rho(x)$ is stochastically larger on $\mathcal{D}_{\text{forget}}$ than on $\mathcal{D}_{\text{retain}}$, we know that $\Pr_{x \sim \mathcal{D}_{\text{retain}}}[\rho(x) > \tau]$ is a strictly decreasing function of $\tau$.

Therefore, there exists a threshold $\tau^* \in [0, 1]$ such that:

$$\Pr_{x \sim \mathcal{D}_{\text{retain}}}[\rho(x) > \tau^*] = \alpha = \Pr_{x \sim \mathcal{D}_{\text{retain}}}[x \in B_{\text{stat}}]$$

This means:

$$\text{SideEffect}(a_{\text{dynamic}}(\tau^*)) = \text{SideEffect}(a_{\text{static}})$$

**Step 3: Show that coverage is greater for the dynamic approach.**   From Theorem 3, we know that thresholding $\rho(x)$ at $\tau^*$ gives the optimal classifier for distinguishing between $\mathcal{D}_{\text{forget}}$ and $\mathcal{D}_{\text{retain}}$ at false positive rate $\alpha$.

More formally, among all sets $A \subseteq \mathcal{X}$ with $\Pr_{x \sim \mathcal{D}_{\text{retain}}}[x \in A] = \alpha$, the set $B_{\text{dyn}}(\tau^*) = \{x : \rho(x) > \tau^*\}$ maximizes $\Pr_{x \sim \mathcal{D}_{\text{forget}}}[x \in A]$.

Since $B_{\text{stat}}$ is one such set with $\Pr_{x \sim \mathcal{D}_{\text{retain}}}[x \in B_{\text{stat}}] = \alpha$, we must have:

$$\Pr_{x \sim \mathcal{D}_{\text{forget}}}[x \in B_{\text{dyn}}(\tau^*)] \geq \Pr_{x \sim \mathcal{D}_{\text{forget}}}[x \in B_{\text{stat}}]$$

Therefore:

$$\text{Coverage}(a_{\text{dynamic}}(\tau^*)) \geq \text{Coverage}(a_{\text{static}})$$

If $\rho(x)$ is *strictly* stochastically larger on $\mathcal{D}_{\text{forget}}$ than on $\mathcal{D}_{\text{retain}}$ (which holds in practice as forget-relevant features activate more frequently on forget-set queries), then this inequality is strict.

We have established that for any static clamping approach, there exists a threshold $\tau^*$ such that the dynamic approach with this threshold achieves the same side-effect on the retain set; and achieves equal or greater coverage on the forget set. This proves that dynamic clamping dominates static clamping in the coverage-side effect trade-off.   □

## F    Distribution of token activations on WMDP-Cyber

Figure 8 plots the distribution of forget-set activated tokens on WMDP-Cyber. The threshold is chosen to control the retain set's false positive rate and we find that $p_{dyn} = 95$ typically separates forget-set queries effectively achieving high recall on $\mathcal{D}_{forget}$. On WMDP-Cyber, DSG successfully transfers from the retain set (WikiText) and forget set to the test query set.

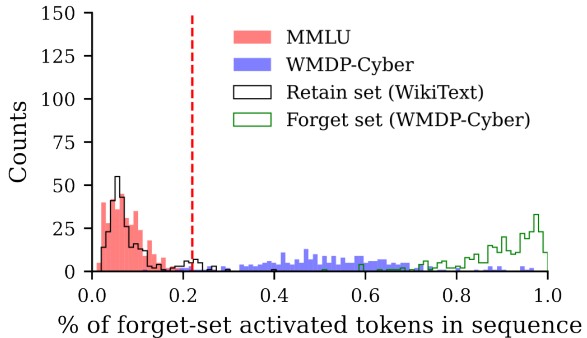

**Figure 8:** Distribution of forget-set activated tokens for WMDP-Cyber. Threshold at the 95th percentile (dashed red line) effectively separates MMLU from WMDP.

## G    Unlearning on WMDP

### G.1    Hyperparameter Details and Model Descriptions for Baselines

To ensure a comprehensive and fair comparison of unlearning methods, we conducted extensive hyperparameter sweeps for each baseline, optimizing for both the effectiveness of knowledge removal and the preservation of model utility. For all gradient-based methods, we experimented with updating parameters in layers 3, 7, and 11 (as recommended in (Li et al., 2024)), as well as all layers. Unless otherwise specified, all experiments used the `google/gemma-2-2b-it` (Lieberum et al., 2024) model.

**Dynamic SAE Guardrails (DSG).**    Our proposed method, DSG, is a non-gradient-based intervention method that selectively removes hazardous knowledge by manipulating SAE feature activations. DSG first identifies a subset of SAE features strongly indicative of the knowledge to be forgotten, based on their differential activation patterns on forget and retain datasets. During inference, DSG employs a dynamic classifier to assess the relevance of input sequences. If a sequence is classified as forget-relevant based on the aggregate activation of selected features, DSG dynamically clamps these features to a negative value. This conditional, sequence-level clamping ensures that intervention is applied only when necessary, minimizing side effects on benign inputs and preserving model utility.

We employed the `gemma-scope-2b-pt-res` SAE (width 16k) applied to layer 3 ($\ell_0$ 142) (Lieberum et al., 2024). The dynamic threshold percentile ($p_{\text{dyn}}$) was fixed at 95. We swept the importance ratio percentile ($p_{\text{ratio}}$), number of selected features, and clamp strength ($c$):

| Hyperparameter | Values Tested |
|---|---|
| Importance Ratio Percentile ($p_{\text{ratio}}$) | 90, 95 |
| Number of Features | 10, 20, 30 |
| Clamp Strength ($c$) | 10, 25, 50, 100, 200, 300, 400, 500 |

**Table 3:** Hyperparameter sweep for Dynamic SAE Guardrails (DSG). Fixed values: $p_{\text{dyn}} = 95$.

The best configurations were: WMDP-Bio ($p_{\text{ratio}} = 95$, features=20, $c = 500$) and WMDP-Cyber ($p_{\text{ratio}} = 90$, features=30, $c = 500$).

**Representation Misdirection for Unlearning (RMU).** RMU (Li et al., 2024) is a gradient-based finetuning method that minimizes a composite loss function to achieve targeted forgetting while preserving model utility. This loss combines a forget loss and a retain loss. The forget loss acts on the model's activations on the forget dataset, increasing their norm in specific directions and making it difficult for later layers to process this information effectively. Simultaneously, the retain loss regularizes the updated model's activations on the retain dataset, encouraging activations to stay close to the original model's activations on benign data.

Key hyperparameters include the steering coefficient, which controls how much the activations are amplified on hazardous data, and the alpha parameter ($\alpha$), which balances utility preservation against knowledge removal. We focus unlearning only on the MLPs, as recommended in Li et al. (2024).

| Hyperparameter | Values Tested |
|---|---|
| Steering Coefficient | 1, 5, 10, 20, 100, 200, 400, 500, 800, 1000 |
| Alpha ($\alpha$) | 0.01, 0.1, 1, 10, 100, 300, 500 |
| Batch Size | 4, 8 |
| Steps | 400, 800 |

**Table 4:** Hyperparameter sweep for RMU. Fixed values: Monitoring Layer ID=3, Learning Rate=5e-6.

The best configuration for WMDP-Bio used steering coefficient 400, alpha 100, monitoring layer 3, learning rate 5e-6, batch size 8, and 400 steps. For WMDP-Cyber, we used steering coefficient 500, alpha 10, monitoring layer 3, and batch size 8 with 400 steps.

**Scalable Remembering and Unlearning unBound (SCRUB).** SCRUB (Kurmanji et al., 2023) employs a student-teacher framework for knowledge distillation-based unlearning. It trains a student model, a clone of the original model, to forget hazardous knowledge under the guidance of the original, frozen teacher model. During forget epochs, SCRUB maximizes the KL divergence between student and teacher logits on the forget dataset. In retain epochs, it minimizes this divergence on the retain dataset, guiding the student to mimic the teacher on benign data.

We swept across values of beta ($\beta$), a weighting factor balancing knowledge distillation and task-specific loss, while fixing alpha ($\alpha$) and gamma ($\gamma$) at 1.0:

| Hyperparameter | Values Tested |
|---|---|
| Beta ($\beta$) | 0.0001, 0.001, 0.01, 0.1, 1, 10 |
| Learning Rate (lr) | 1e-4, 1e-5, 5e-6 |
| Batch Size | 4, 8 |
| Steps | 400, 800 |

**Table 5:** Hyperparameter sweep for SCRUB. Fixed values: $\alpha = 1.0$, $\gamma = 1.0$, KL Temperature=2.0.

The best configuration for WMDP-Bio used beta 0.01, learning rate 5e-6, batch size 8, and 400 maximum batches. For WMDP-Cyber, we used beta 0.1, learning rate 1e-5, batch size 8, and 400 maximum batches.

**Selective Synaptic Dampening (SSD).** SSD (Foster et al., 2024) identifies and dampens parameters more important for the forget set than the retain set. It adapts a method originally developed for image classification to language modeling by modifying the loss function to use log-perplexity. SSD calculates parameter importance scores based on gradients observed for both forget and retain datasets, then applies a dampening factor to parameters with higher importance for the forget dataset.

We performed a grid search spanning dampening thresholds and constants:

| Hyperparameter | Values Tested |
|---|---|
| Threshold | 0.1, 0.25, 0.5, 1, 2.5, 5 |
| Dampening Constant | 1e-5, 1e-4, 1e-3, 1e-2, 1e-1, 1 |

**Table 6:** Hyperparameter sweep for Selective Synaptic Dampening (SSD).

The optimal configuration for WMDP-Bio used threshold 0.5 and dampening constant 1e-3. For WMDP-Cyber, we used threshold 1.0 and dampening constant 1e-2.

**Static SAE Clamping (Farrell et al.).** This non-gradient-based approach (Farrell et al., 2024) identifies salient SAE features and statically clamps their activations during inference to remove unwanted knowledge. Unlike our dynamic approach, this method applies feature clamping universally to all inputs whenever a selected feature activates, rather than conditionally based on sequence-level classification.

We varied the retain threshold, multiplier (clamp value), and number of features:

| Hyperparameter | Values Tested |
|---|---|
| Retain Threshold | 0.01, 0.001, 0.005, 0.1, 1 |
| Multiplier (Clamp Value) | 10, 25, 50, 100, 200, 500 |
| Number of Features | 5, 10, 20, 30, 50 |

**Table 7:** Hyperparameter sweep for Static SAE Clamping. Fixed value: Sequence Length=1024.

The best configurations were: WMDP-Bio (retain threshold=0.01, multiplier=200, features=5) and WMDP-Cyber (retain threshold=0.005, multiplier=500, features=10).

**Gradient Ascent (GA).** GA (Jang et al., 2023) is a finetuning-based unlearning method that directly minimizes the likelihood of correct predictions on the forget dataset using gradient ascent. In contrast to standard finetuning which employs gradient descent, GA utilizes gradient ascent to maximize the cross-entropy loss on the forget dataset, pushing parameters in directions that increase prediction error on the targeted data.

We varied the learning rate and beta ($\beta$), the retain loss weight:

| Hyperparameter | Values Tested |
|---|---|
| Learning Rate (lr) | 1e-5, 5e-5 |
| Beta (Retain Loss Weight) | 0.01, 0.1, 1.0, 5.0, 10.0 |

**Table 8:** Hyperparameter sweep for Gradient Ascent. Fixed values: Gamma=1.0, Batch Size=8, Steps=400.

We explored both with and without retain data configurations. The best setting for WMDP-Bio used learning rate 1e-5 with beta 1.0. For WMDP-Cyber, we used learning rate 1e-5 with beta 0.1.

**Negative Preference Optimization (NPO).** NPO (Zhang et al., 2024) adapts preference optimization techniques to treat the forget set as negative examples. It reframes unlearning as preference learning, optimizing the model to assign lower likelihood to the forget set. The beta parameter controls the extent to which the unlearned model's output distribution can diverge from the original model. To mitigate utility degradation and preserve performance on benign data, NPO can be regularized using two distinct retain loss types: Negative Log-Likelihood (NLL) and Kullback-Leibler (KL) divergence. NLL minimization directly encourages the model to maintain high probabilities for correct tokens in the retain set, calculated as the negative sum of log probabilities assigned to ground truth tokens. KL divergence minimization encourages the probability distribution of the unlearned model to remain close to that of the original model on retain set inputs, measured as the information lost when approximating the original model's distribution with the unlearned model's distribution.

We tested NPO with various configurations:

| Hyperparameter | Values Tested |
|---|---|
| Alpha (Retain Loss Weight) | 0.01, 0.1, 1.0 |
| Beta (Temperature Parameter) | 0.1, 1.0 |
| Retain Loss Type | NLL, KL |

**Table 9:** Hyperparameter sweep for NPO. Fixed values: Gamma=1.0, Learning Rate=1e-5, Batch Size=8, Steps=400.

The optimal settings for WMDP-Bio used alpha 0.1, beta 0.1, and KL divergence as the retain loss type and WMDP-Cyber used alpha 1.0, beta 0.1, and KL divergence as the retain loss type.

For all methods, we selected configurations that minimized WMDP accuracy while maintaining at least 99% of the original model's MMLU accuracy.

**Compute.** All finetuning and inference was performed on 4 A6000 GPUs in under a day.

## G.2 Results on WMDP-Cyber

Table 10 shows the performance of various unlearning baselines on WMDP-Cyber dataset. RMU is less effective on WMDP-Cyber (88.00%), likely due to the data inefficiency of gradient-based methods on the smaller cyber forget set.

| Method | WMDP Cyber ($\downarrow$) | MMLU ($\uparrow$) | | | | | MT ($\uparrow$) |
|---|---|---|---|---|---|---|---|
| | | HS Hist | C. Bio | HS Geo | H. Aging | All | |
| Target $\mathcal{M}$ | 100.00 | 100.00 | 100.00 | 100.00 | 100.00 | 100.00 | 7.36 |
| GA | 98.91 | 98.15 | 100.0 | 100.0 | 100.0 | 99.46 | 7.39 |
| NPO | 96.36 | 100.0 | 100.0 | 100.0 | 100.0 | 100.0 | 7.18 |
| SSD | 98.91 | 100.00 | 100.00 | 98.08 | 98.81 | 99.19 | 7.25 |
| SCRUB | 97.82 | 99.07 | 100.00 | 100.00 | 98.81 | 99.46 | 6.51 |
| Farrell et al. | 52.73 | 99.07 | 100.00 | 100.00 | 97.62 | 99.19 | 7.39 |
| RMU | 88.00 | 99.07 | 100.00 | 99.04 | 98.81 | 99.19 | 7.28 |
| **DSG (Ours)** | 26.74 | 99.07 | 100.00 | 100.00 | 100.00 | 99.73 | 7.66 |

**Table 10:** Unlearning performance on WMDP-Cyber. All represents the average MMLU score. MT-Bench scores show 0.13 variance across 5 runs. DSG shows superior unlearning effectiveness compared to other baselines while maintaining high MMLU performance.

## G.3 MT-Bench Evaluation Details

To measure the impact of unlearning on the model's general conversational abilities and fluency, we utilized the MT-Bench benchmark (Zheng et al., 2023). Specifically, we report the average score across two conversational turns (the two-turn average score), which provides a measure of multi-turn conversational quality. Following standard MT-Bench protocol, evaluations were conducted using GPT-4 (Achiam et al., 2023) as the judge to score the model's responses. To ensure the robustness of these fluency assessments, each model configuration reported in Section 4.1 was evaluated 5 times using MT-Bench. The mean scores presented in Table 1 and Table 10 reflect the average performance across these runs, and the standard deviation across the 5 runs is noted in the respective table captions (0.16 for WMDP-Bio results, 0.13 for WMDP-Cyber results). Higher MT-Bench scores indicate better preservation of general conversational capabilities after the unlearning procedure.

# H  Unlearning on MUSE

## H.1  Hyperparameter Details and Model Descriptions for Baselines

We provide implementation details for the baseline unlearning methods evaluated in our experiments.

**Gradient Ascent (GA).**  GA maximizes the loss on the forget set, directly opposing the standard training objective to push the model away from the forget data's distribution (Jang et al., 2023). While straightforward, it often leads to catastrophic forgetting of general knowledge.

**Gradient Difference (GradDiff).**  GradDiff balances competing objectives by maximizing the loss on the forget set while minimizing the loss on the retain set (Liu et al., 2022). Despite this approach, GradDiff struggles to find an optimal trade-off, resulting in either over- or under-unlearning.

**Negative Preference Optimization (NPO).**  NPO reframes unlearning within a preference learning framework, treating the forget set as negative preference data by adapting the Direct Preference Optimization objective (Zhang et al., 2024). We use NPO with KL Divergence Minimization that augments NPO with a KL divergence term to preserve utility by minimizing distributional shift on benign data.

**Simplified NPO (SimNPO).**  A computationally efficient variant of NPO that simplifies the optimization process while retaining core principles of negative preference learning (Fan et al., 2024). SimNPO trades some unlearning effectiveness for faster processing.

**Representation Misdirection for Unlearning (RMU).**  RMU injects targeted noise into specific layers to disrupt the model's ability to process information related to the forget set (Li et al., 2024). Its effectiveness depends heavily on precise noise targeting and hyperparameter tuning. We injected noise in the 7th layer for both News and Books.

For all finetuning-based baselines (GA, GradDiff, NPO, SimNPO, RMU), we used AdamW optimizer with a learning rate of 1e-5 and batch size of 32. We finetuned all parameters in the model. The optimal checkpoint for each method was determined by selecting the first epoch (within 10 epochs) where the unlearned model's utility on the retain set fell below 90% that of the target model. Table 11 summarizes the optimal epochs or $\alpha$ values for each method on both datasets.

| Unlearning Method | NEWS | BOOKS |
|---|---|---|
| GA | epoch 1 | epoch 1 |
| GradDiff | epoch 2 | epoch 3 |
| NPO | epoch 8 | epoch 10 |
| SimNPO | epoch 10 | epoch 10 |
| RMU | epoch 9 | epoch 10 |

**Table 11:** Optimal epochs for baseline unlearning methods on MUSE benchmark, determined by utility-based stopping criteria.

All finetuning and inference was performed on 4 A6000 GPUs in under a day.

## H.2  Sequential Unlearning Strategies for DSG

In real-world scenarios, unlearning requests often arrive sequentially over time. An effective unlearning method must be able to handle multiple, successive requests without significant degradation in performance or utility. In Section 3, we evaluated DSG's performance under sequential unlearning using the MUSE benchmark (Shi et al., 2024) with four disjoint folds of the NEWS corpus. We implemented and compared two strategies for adapting DSG to this sequential setting, referred to as DSG$_{\text{all}}$ and DSG$_{\text{union}}$. Both strategies leverage the core DSG mechanisms of feature selection and dynamic thresholding but differ in how they aggregate information across multiple unlearning requests.

**Setup.** Let $k = 1, 2, \ldots, K$ index the sequential unlearning requests. Each request $k$ introduces a new forget dataset $D_{F,k}$. We assume the retain dataset $D_R$ remains constant throughout the process. The goal at step $k$ is to produce an unlearned model that effectively forgets the cumulative forget data $\mathcal{D}_{F,k}^{\text{cumul}} = \cup_{i=1}^{k} D_{F,i}$ while preserving utility evaluated on $D_R$. Let $n_{\text{feats}}$ be the desired number of features to select at each relevant stage.

**Strategy 1: DSG$_{\text{all}}$ (Cumulative Score Update)** This strategy treats the sequential unlearning problem as equivalent to unlearning a single, growing forget set $\mathcal{D}_{F,k}^{\text{cumul}}$ at each step $k$. It maintains cumulative statistics required for calculating the feature importance scores.

- **Cumulative Statistics:** At step $k$, we need the aggregate sum of squared activations and the total number of tokens for all forget data seen so far. Let $A_{F,i}^2(j) = \sum_{x \in D_{F,i}} \sum_{t=1}^{|x|} [f_j(\mathbf{h}_{x,t})]^2$ be the sum of squared activations for feature $j$ on dataset $D_{F,i}$, and $N_{F,i} = \sum_{x \in D_{F,i}} |x|$ be the total number of tokens in $D_{F,i}$. The cumulative sums at step $k$ are:

$$\Sigma_{F,k}^2(j) = \sum_{i=1}^{k} A_{F,i}^2(j)$$

$$N_{F,k}^{\text{cumul}} = \sum_{i=1}^{k} N_{F,i}$$

These sums can be updated incrementally as each new $D_{F,k}$ arrives, without needing to store all previous datasets. The retain set statistics ($A_R^2(j)$ and $N_R$) are computed once from $D_R$.

- **Score Calculation:** The importance scores are calculated using the cumulative statistics:

$$\text{forget\_score}_{\text{all},k}(j) = \frac{\Sigma_{F,k}^2(j)}{N_{F,k}^{\text{cumul}}}$$

$$\text{retain\_score}(j) = \frac{A_R^2(j)}{N_R} \quad \text{(constant across } k\text{)}$$

$$\text{imp\_ratio}_{\text{all},k}(j) = \frac{\text{forget\_score}_{\text{all},k}(j)}{\max\{\text{retain\_score}(j), \varepsilon\}}$$

- **Feature Selection:** Using $\text{imp\_ratio}_{\text{all},k}(j)$ and $\text{forget\_score}_{\text{all},k}(j)$, select the feature set $S_{\text{all},k}$ containing the top $n_{\text{feats}}$ features, following the procedure in Algorithm 1 (filtering by percentile $p_{\text{ratio}}$ and ranking by forget score).
- **Dynamic Threshold and Intervention:** Calculate the activation statistic $\rho_{\text{all},k}(x)$ as $\rho_{\text{all},k}(x) = (1/|x|) \sum_t \mathbf{1}[\exists j \in S_{\text{all},k} : f_j(\mathbf{h}_t) > 0]$. Calibrate the dynamic threshold $\tau_{\text{all},k} = \text{Percentile}(\{\rho_{\text{all},k}(x)\}_{x \in D_R}, p_{\text{dyn}})$. Apply conditional clamping using $S_{\text{all},k}$ and $\tau_{\text{all},k}$ during inference.

DSG$_{\text{all}}$ aims for the most accurate representation of feature importance with respect to all forgotten data combined.

**Strategy 2: DSG$_{\text{union}}$ (Union of Feature Sets)** This strategy selects features based on each individual forget request $D_{F,k}$ and then uses the union of these feature sets for intervention.

- **Independent Score Calculation:** At step $k$, calculate importance scores using only the current forget set $D_{F,k}$ and the retain set $D_R$:

$$\text{forget\_score}_{\text{indep},k}(j) = \frac{A_{F,k}^2(j)}{N_{F,k}}$$

$$\text{retain\_score}(j) = \frac{A_R^2(j)}{N_R}$$

$$\text{imp\_ratio}_{\text{indep},k}(j) = \frac{\text{forget\_score}_{\text{indep},k}(j)}{\max\{\text{retain\_score}(j), \varepsilon\}}$$

- **Independent Feature Selection:** Select the feature set $S_{\text{indep},k}$ containing the top $n_{\text{feats}}$ features based on $\texttt{imp\_ratio}_{\text{indep},k}(j)$ and $\texttt{forget\_score}_{\text{indep},k}(j)$.
- **Union Set Formation:** Maintain the cumulative union of feature sets identified at each step:
$$S_{\text{union},k} = S_{\text{union},k-1} \cup S_{\text{indep},k} \quad (\text{with } S_{\text{union},0} = \varnothing)$$
The size of $S_{\text{union},k}$ may grow beyond $n_{\text{feats}}$.
- **Dynamic Threshold and Intervention:** Calculate the activation statistic $\rho_{\text{union},k}(x) = \frac{1}{|x|} \sum_t \mathbf{1}[\exists j \in S_{\text{union},k} : f_j(\mathbf{h}_t) > 0]$. Calibrate the dynamic threshold $\tau_{\text{union},k} = \text{Percentile}(\{\rho_{\text{union},k}(x)\}_{x \in D_R}, p_{\text{dyn}})$. Apply conditional clamping using $S_{\text{union},k}$ and $\tau_{\text{union},k}$ during inference.

$\text{DSG}_{\text{union}}$ ensures that features deemed important for any past forget request are considered for intervention, potentially capturing a broader range of forget-related concepts but possibly leading to a larger intervention set over time.

**Result.** As reported in the main text (Figure 4(b)), both $\text{DSG}_{\text{all}}$ and $\text{DSG}_{\text{union}}$ demonstrated strong and stable performance across the four sequential unlearning requests on the MUSE benchmark, significantly outperforming gradient-based methods which showed rapid degradation.

# I  Relearning attack

## I.1  Superficial Alignment Hypothesis

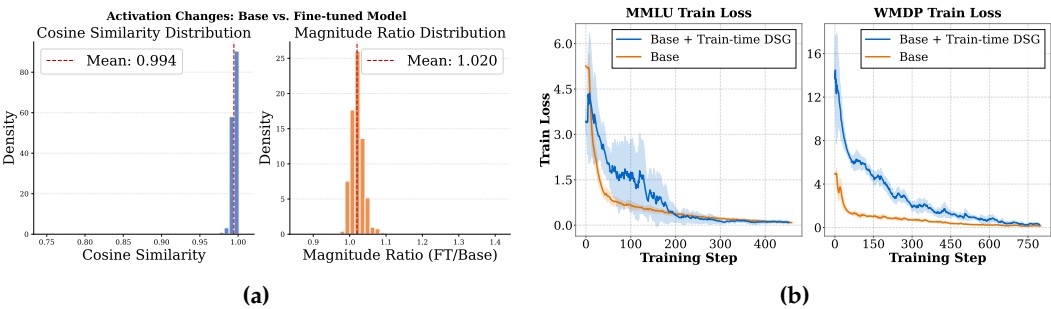

**Figure 9:** (a) Distribution of activation cosine similarity and activation magnitude ratio between Base and Finetuned models. Finetuning does not significantly change the underlying activation space. (b) Train loss when finetuning Base model and Base+SAE model on WMDP and MMLU. Loss on WMDP for the BASE+SAE model is significantly higher than on MMLU.

The resistance of DSG to relearning attacks can be understood through the lens of the Superficial Alignment Hypothesis (Zhou et al., 2023), which posits that a model's activation geometry is established during pretraining and remains relatively stable during subsequent finetuning. We provide empirical evidence supporting this hypothesis in Figure 9a, which presents the distribution of activation cosine similarities and magnitude ratios between the base and finetuned models.

The concentration of cosine similarity values near 1.0 indicates that finetuning preserves the directional information in the activation space, with minimal rotational changes. Similarly, the activation magnitude ratios cluster tightly around 1.0, demonstrating that the scale of activations remains largely unchanged during finetuning. These findings align with previous research suggesting that while weights may change substantially during finetuning, the underlying activation patterns and geometry remain remarkably stable.

This stability of activation geometry is the basis for DSG's effectiveness against relearning attacks. By operating directly on these stable activation patterns rather than weights, DSG establishes a more durable defense mechanism that persists even when adversaries attempt to modify the model's weights through finetuning.

### I.2 Train-time DSG Details

Beyond applying DSG only at inference (Test-time DSG), we explore integrating it directly into the finetuning process itself to further enhance resistance against relearning attacks. This approach, termed **Train-time DSG**, applies the standard DSG logic during each forward pass of the finetuning/relearning phase.

Specifically, during finetuning on a potentially adversarial dataset (like the forget set itself in a relearning attack scenario), Train-time DSG operates as follows:

1. For each input sequence $x$ in a training batch, compute the hidden states $h_t$ and corresponding SAE feature activations $f_j(h_t)$.
2. Calculate the statistic $\rho(x)$ based on the pre-selected forget feature set $S_{\text{nfeats}}$.
3. Classify the sequence using the dynamic threshold $\tau$: $C(x) = \mathbf{1}[\rho(x) > \tau]$.
4. **Conditional Clamping:** If $C(x) = 1$ (forget-relevant), modify the activations for features $j \in S_{\text{nfeats}}$ by setting $f'_j(h_t) = -c$ for all tokens $t$. Otherwise, $f'_j(h_t) = f_j(h_t)$. These potentially modified activations $f'$ are then used for the reconstruction $\hat{h}'$ and subsequent layers of the LLM.
5. The final loss (e.g., cross-entropy on the relearning task) is computed based on the LLM's output derived from these potentially clamped activations.
6. **Gradient Blocking:** During the backward pass, gradients flow back through the model as usual. However, for any feature activation $f_j(h_t)$ that was clamped to $-c$, the gradient of the loss with respect to the upstream components (that produced $h_t$) through that specific feature pathway is effectively blocked. Setting the activation to a constant $-c$ detaches it from the upstream computations for the purpose of gradient calculation via that feature's contribution path. This is conceptually akin to applying a `stop_gradient` operation specifically on the clamped feature activations.

During this finetuning process, the parameters of the SAE itself (encoder $W_{\text{enc}}, b_{\text{enc}}$ and decoder $W_{\text{dec}}, b_{\text{dec}}$) are kept **frozen**. This prevents the SAE from adapting to circumvent the clamping intervention.

This dynamic gradient blocking prevents the finetuning process from easily undoing the unlearning effect by simply adjusting weights to reactivate the specific features in $S_{\text{nfeats}}$ that carry the forget-set information. When the model attempts to minimize loss on forget-set examples by utilizing these features, Train-time DSG clamps them and blocks the relevant gradient signal. This forces the model, if it attempts to relearn, to find potentially much less direct or alternative pathways through other features or model components. This difficulty in relearning via the original pathways contributes to the significantly higher training loss observed on WMDP-Bio when finetuning with Train-time DSG active, as seen in Figure 9b.

### I.3 Tamper-Resistant Safeguards

DSG functions as a tamper-resistant safeguard during finetuning by effectively filtering gradients that would otherwise enable the model to relearn forgotten knowledge. Figure 9b demonstrates this mechanism quantitatively, showing the training loss profiles when finetuning the base model and the base model with DSG active on both WMDP-Bio (forget set) and MMLU (retain set) datasets.

When DSG is active during finetuning, we observe significantly elevated training loss values on WMDP-Bio compared to MMLU. This marked difference in loss profiles indicates that DSG selectively impedes the model from reducing loss on forget set content while allowing normal optimization on retain set content. This selective gradient filtering creates an effective barrier against relearning targeted information.

The mechanism works because during finetuning, DSG constantly monitors activations and applies clamping whenever forget-relevant features are activated above the dynamic threshold. This intervention disrupts the gradient flow for targeted concepts, requiring the model to develop entirely new processing pathways rather than simply recovering previously established connections. This *rewiring* requirement explains the delayed recovery

pattern observed in the main relearning experiments, where performance remains near random for approximately six epochs before beginning to increase.

### I.4 Relearning Attack at Learning Rate 1e-6

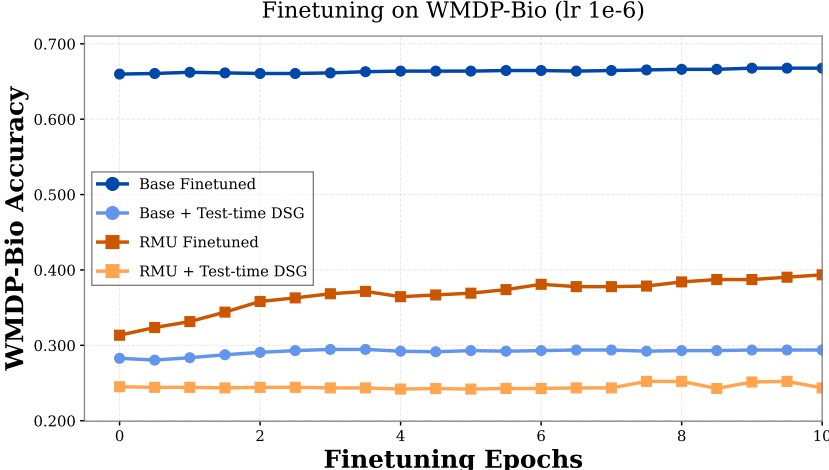

**Figure 10:** Relearning attack performance with reduced learning rate (1e-6). All configurations show minimal performance changes across finetuning epochs, demonstrating that relearning attack efficacy is strongly dependent on learning rate.

To investigate the impact of learning rate on relearning attack efficacy, we conducted a supplementary analysis using a reduced learning rate of 1e-6 (compared to 1e-5 in the main experiments). Figure 10 presents WMDP-Bio accuracy across finetuning epochs for all configurations under this reduced learning rate condition.

The results demonstrate minimal performance changes across all configurations throughout the finetuning process. This stability indicates that relearning attack efficacy is strongly dependent on learning rate, with lower rates substantially limiting the model's ability to recover forgotten knowledge. This finding has important implications for practical deployment scenarios, suggesting that implementing learning rate constraints on model access APIs could serve as an additional defense layer against relearning attacks.

### I.5 Relearning Hyperparameters

For the relearning experiments, we used the RMU unlearned model as described in Section 4.1, with RMU hyperparameters set to steering coefficient 400, alpha 100, monitoring layer 3, AdamW optimizer, learning rate 5e-6, batch size 8, and 400 steps. For DSG configurations, we employed the optimal parameters identified in our WMDP-Bio experiments: importance ratio percentile ($p_{\text{ratio}}$) of 95, feature count of 20, and clamp strength ($c$) of 500 for both test-time and train-time DSG interventions. The dynamic threshold percentile ($p_{\text{dyn}}$) was maintained at 95, consistent with our main experiments.

All finetuning and inference was performed on 2 A100 GPUs in under a day.

## J Data Efficiency and Zero-shot Capabilities

### J.1 Hyperparameters

**Data Efficiency**   For the data efficiency experiments, we maintain consistent hyperparameter settings across all data subsets to isolate the impact of dataset size. We use the optimal DSG configuration identified for WMDP-Bio with 100% data, as shown in Table 12.

| Parameter | Value |
|---|---|
| SAE | gemma-scope-2b-pt-res SAE (width 16k) |
| SAE layer | Layer 3, $\ell_0$ 142 |
| Importance ratio percentile ($p_{\text{ratio}}$) | 95 |
| Dynamic threshold percentile ($p_{\text{dyn}}$) | 95 |
| Number of selected features | 20 |
| Clamp strength ($c$) | 500 |

**Table 12:** DSG hyperparameters for data efficiency experiments

For RMU comparisons, we evaluate two approaches: (1) maintaining the same number of training steps (400) across all data subsets, and (2) completing one full epoch over each dataset subset. Maintaining the same number of training steps produced a better Pareto front. We select the model with lower WMDP accuracy for each subset. The base RMU configuration for WMDP-Bio is presented in Table 13.

| Parameter | Value |
|---|---|
| Steering coefficient | 400 |
| Alpha ($\alpha$) | 100 |
| Monitoring layer | 3 |
| Learning rate | 5e-6 |
| Parameter subset | MLP layers only |

**Table 13:** RMU hyperparameters for data efficiency experiments

**Zero-shot** For zero-shot experiments, we vary only the dynamic threshold $\tau$ (as no retain set is available for calibration) while keeping all other hyperparameters fixed at their optimal values for each task, as shown in Table 14.

| Parameter | WMDP-Bio | WMDP-Cyber |
|---|---|---|
| SAE layer | Layer 3, $\ell_0$ 59 | Layer 3, $\ell_0$ 59 |
| Importance ratio percentile ($p_{\text{ratio}}$) | 95 | 90 |
| Feature selection | 20 features | 20 features via |
| Clamp strength ($c$) | 500 | 500 |
| $\tau$ range tested | 0.1 to 0.9 (increments of 0.1) | 0.1 to 0.9 (increments of 0.1) |
| Optimal $\tau$ | 0.6 | 0.2 |

**Table 14:** Hyperparameters for zero-shot experiments

The optimal thresholds were determined to be $\tau = 0.6$ for WMDP-Bio and $\tau = 0.2$ for WMDP-Cyber, as shown in Figure 6B.

# K  Ablations

This appendix provides comprehensive details on our ablation studies for DSG. We analyze each component's contribution to overall performance and explore sensitivity to various hyperparameters.

## K.1  Additional Ablations

**DSG Clamp Strength $c$.** The clamping parameter $c$ determines the magnitude of intervention applied to selected SAE features. As shown in Figure 11, WMDP-Bio accuracy drops significantly at modest clamp values ($c = 25$), reaching near-optimal unlearning performance, while MMLU accuracy remains above 99% for configurations with 10-20 features. For these optimal feature counts, performance remains remarkably stable across a wide range of clamp strengths ($100 \le c \le 500$), demonstrating DSG's robustness to this parameter. By contrast Farrell et al. (2024) exhibit greater sensitivity to clamp values, as seen in Figure 7A.

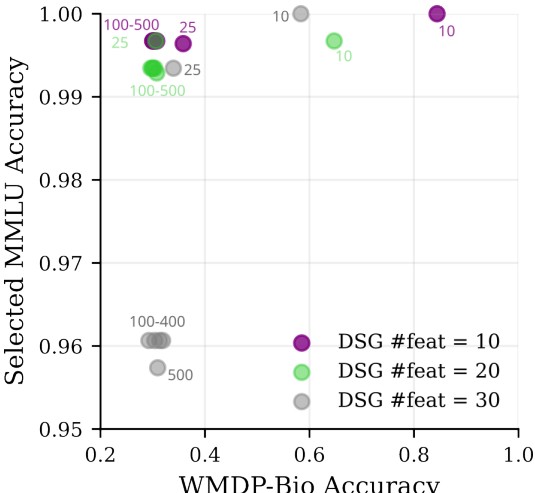

**Figure 11:** Effect of clamp strength $c$ on DSG performance across different feature counts. MMLU accuracy (solid lines) remains consistently high ($> 99\%$) for 10-20 features across all clamp values, while WMDP-Bio accuracy (dashed lines) drops sharply even at modest clamp strengths ($c = 25$). This demonstrates DSG's ability to effectively remove targeted knowledge while preserving general model capabilities with minimal parameter sensitivity.

**DSG Number of Features.**  Across experiments, the number of features selected during percentile-based feature selection represents a critical balance between coverage and precision. Selecting too few features may result in insufficient removal of forget-set information, as some forget-set inputs might not activate the limited feature set strongly enough to trigger intervention. Conversely, selecting too many features increases the risk of including noisy features selected using importance scoring or less discriminative features that activate on retain-set samples, potentially causing false positive detections and reducing model utility.

Our experiments consistently show that 20 features provides an optimal balance for both WMDP-Bio and WMDP-Cyber domains. Configurations with 10 features occasionally show reduced unlearning effectiveness despite good utility preservation, while 30-feature configurations begin to impact retain-set performance at higher clamp strengths. The precise optimal feature count may vary by domain and dataset characteristics as well as SAE width, but the overall pattern of diminishing returns with increased feature counts remains consistent.

**Choice of Activation Statistic: Percentage vs. Raw Count.**  DSG's dynamic classification uses a sequence-level statistic derived from forget-feature ($S_{n_{\text{feats}}}$) activations. We compared two statistics: (1) Percentage-based ($\rho$), the fraction of tokens where any $j \in S_{n_{\text{feats}}}$ activates ($f_j(\mathbf{h}_t) > 0$):

$$\rho(x) = \frac{1}{|x|} \sum_{t=1}^{|x|} \mathbf{1}[\exists j \in S_{n_{\text{feats}}} : f_j(\mathbf{h}_t) > 0]$$

and (2) Raw count-based ($\rho_{\text{raw}}$), the absolute number of such tokens:

$$\rho_{\text{raw}}(x) = \sum_{t=1}^{|x|} \mathbf{1}[\exists j \in S_{n_{\text{feats}}} : f_j(\mathbf{h}_t) > 0]$$

Effective dynamic thresholding (calibrated on WikiText) requires low distributional distance (Total Variation Distance, TVD) between retain sets (WikiText vs. MMLU) for generalization, and high TVD between retain and forget sets (WikiText vs. WMDP) for discrimination (Figure 12).

Empirically, $\rho$ performs significantly better. For WMDP-Bio: (1) Retain alignment (WikiText vs. MMLU): TVD($\rho$) = $0.38 \pm 0.03$ vs. TVD($\rho_{\text{raw}}$) = $0.88 \pm 0.01$, indicating $\rho$ generalizes better across retain sets. (2) Retain/Forget separation (WikiText vs. WMDP-Bio): TVD($\rho$) =

$0.90 \pm 0.02$ vs. $\mathrm{TVD}(\rho_{\mathrm{raw}}) = 0.41 \pm 0.03$, showing $\rho$ discriminates more effectively. Similar results hold for WMDP-Cyber (Figure 12).

The percentage-based statistic $\rho$ outperforms $\rho_{\mathrm{raw}}$ due to its inherent normalization. Raw counts ($\rho_{\mathrm{raw}}$) are confounded by sequence length, whereas $\rho$ measures activation density, providing a length-invariant signal. This normalization improves both generalization across retain data and discrimination from forget data, making $\rho$ the more robust choice for DSG.

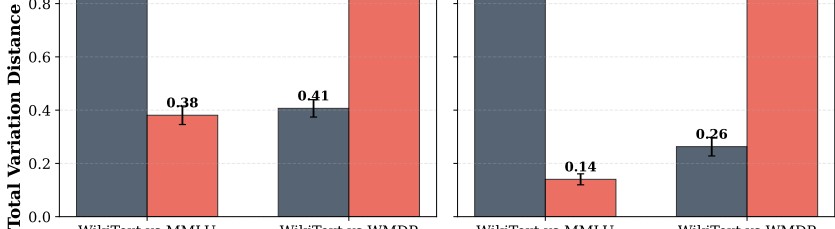

**Figure 12:** Total Variation Distance (TVD) between WikiText and benchmark datasets using percentage-based ($\rho$) vs. raw count-based ($\rho_{\mathrm{raw}}$) metrics. Lower TVD between WikiText and MMLU indicates better alignment of retain sets, while higher TVD between WikiText and WMDP indicates better separation between retain and forget distributions. Percentage-based metrics consistently outperform raw counts on both measures across all benchmarks.

### K.2 Ablations Hyperparameter Details

Our ablation studies used the following hyperparameter configurations:

**Clamp Strength and Feature Count.** We evaluated DSG performance with feature counts of 10, 20, and 30, across clamp strengths $c \in \{10, 25, 50, 100, 200, 300, 400, 500\}$ and $p_{\mathrm{ratio}} = 95$.

**Feature Selection Comparison.** To compare our percentile-based approach with Farrell et al. (2024), we tested both methods using 20 and 30 features, with clamp values in the range [10-500]. We set $p_{\mathrm{ratio}} = 95$ for DSG and used the recommended threshold of 0.01 for Farrell et al. (2024).

**Dynamic Threshold.** We varied $p_{\mathrm{dyn}}$ from 60 to 97 using 20 and 30 features with $c = 500$ to examine the impact of threshold selection on the forget-retain trade-off.

**Importance Ratio Threshold.** We tested $p_{\mathrm{ratio}}$ values from 75 to 95 using 20 and 30 features with $c = 500$ to assess feature selection stringency effects.

**Activation Metrics.** For comparing percentage vs. raw count metrics, we applied bootstrap resampling with 1000 iterations, using Kernel Density Estimation to compute robust TVD estimates between WikiText and test set distributions.

## L Computational Cost (Inference Latency)

A practical consideration for deploying unlearning methods is their impact on inference speed. We evaluated the latency introduced by DSG compared to the original model and a static clamping baseline (Farrell et al., 2024).

Interventions using SAEs inherently introduce some latency compared to the original LLM without the SAE. This overhead stems from two main sources: **(1) The baseline**

**cost of the SAE's forward pass**, which involves matrix multiplications for both encoding ($\mathbf{z} = \sigma(\mathbf{W}_{\text{enc}}\mathbf{h} + \mathbf{b}_{\text{enc}})$) and decoding ($\hat{\mathbf{h}} = \mathbf{W}_{\text{dec}}\mathbf{z} + \mathbf{b}_{\text{dec}}$), scaling with the SAE's width ($d_{\text{sae}}$); and **(2) The cost of the specific intervention logic** applied to the SAE features.

For DSG, this intervention logic involves two main steps beyond the standard SAE pass: (a) calculating the $\rho(x)$ statistic (fraction of forget-activated tokens) across the sequence's activations, and (b) conditionally applying the clamping intervention based on the $\rho(x) > \tau$ comparison.

Table 15 presents the mean inference times (in seconds) and standard deviations over 100 samples (batch size 1) for processing sequences of varying lengths (256, 512, 1024 tokens). These measurements were performed using the `google/gemma-2-2b-it` model (Lieberum et al., 2024) and with the `gemma-scope-2b-pt-res` SAE (width 16k, applied at layer 3, $\ell_0$ 142) (Lieberum et al., 2024) on a single A6000 GPU.

As shown in Table 15, the **total combined overhead** (SAE matrix multiplications + intervention logic) introduced by both static clamping and DSG is **minimal**. Specifically, DSG increases latency by about 5% (ranging from approximately 3.6% to 7.3%) over the original model across the tested sequence lengths. Importantly, the additional overhead incurred by DSG's dynamic classification logic (calculating $\rho(x)$ and thresholding) compared to simple static clamping is negligible indicating that the primary source of the observed latency increase relative to the base LLM is the SAE's own forward pass.

| Seq Length | Original Model (s) | Static Clamping (s) | Dynamic Clamping (DSG) (s) |
|---|---|---|---|
| 256 tokens | $0.0872 \pm 0.0098$ | $0.0933 \pm 0.0091$ (+7.0%) | $0.0936 \pm 0.0090$ (+7.3%) |
| 512 tokens | $0.1618 \pm 0.0061$ | $0.1659 \pm 0.0029$ (+2.5%) | $0.1676 \pm 0.0047$ (+3.6%) |
| 1024 tokens | $0.3300 \pm 0.0081$ | $0.3403 \pm 0.0083$ (+3.1%) | $0.3420 \pm 0.0081$ (+3.6%) |

**Table 15:** Comparison of Inference Latency Across Sequence Lengths for `gemma-2-2b-it` with `gemma-scope-2b-pt-res` SAE. Data reported as mean $\pm$ std over 100 samples on a single A6000 GPU. Percentage increase relative to the Original Model shown in parentheses.

While DSG introduces this slight inference overhead, it is important to consider the broader computational context. Gradient-based unlearning methods require computationally intensive finetuning processes involving backward passes through the model for each unlearning request. In contrast, DSG's unlearning cost primarily involves a one-time computation of activation statistics (which can be amortized across many uses) and the **minimal**, constant inference-time overhead detailed above.

Therefore, DSG offers a highly efficient alternative for unlearning, achieving state-of-the-art forgetting effectiveness and utility preservation with only a marginal increase in inference latency. This makes it particularly attractive for scenarios requiring frequent or sequential unlearning operations where the cost of repeated gradient-based finetuning would be prohibitive.

## M  Feature Interpretability

A key strength of Dynamic SAE Guardrails (DSG) is **interpretable unlearning**, especially in zero-shot scenarios where domain-specific data is absent. To demonstrate this, we used Neuronpedia API's *search by SAE* (Lin, 2023) to directly identify Sparse Autoencoder (SAE) features relevant to biosecurity and cybersecurity hazards. For WMDP-Bio and WMDP-Cyber, "Biology" and "Cybersecurity" queries retrieved the top 20 feature IDs from the `gemma-scope-2b-pt-res` SAE (width 16k) (Lieberum et al., 2024) applied to `gemma-2-2b-it` layer 3 ($\ell_0$ 59).

Table 16 shows the **semantic alignment** of these zero-shot features with the targeted knowledge. Listing the top 20 SAE feature IDs for both domains, alongside Neuronpedia interpretations, the table shows features for "Biology" consistently described with terms like "biological processes", "cellular functions", and "genetics"—core concepts of biosecurity risks. Similarly, "Cybersecurity" features are linked to "cyber threats", "digital security", and "encryption," reflecting cybersecurity risks in WMDP-Cyber. This highlights SAEs' ability to extract topically precise features, even without task-specific data.

Figure 13 further illustrates this, visualizing activations on WMDP-Bio and WMDP-Cyber forget set sequences. Figure 13A (WMDP-Bio) shows activations for IDs 373 and 10933 clustering around biological terms like "bacteria", "cellular", and "infection" while Figure 13B (WMDP-Cyber, IDs 15286 and 2905) shows clusters around cybersecurity terms like "encryption", "data", and "security."

These examples and Table 16 show that **zero-shot SAE feature selection captures semantically rich, domain-relevant concepts associated with hazardous knowledge**. This interpretability is **prescriptive for unlearning**: by targeting these topically coherent features, DSG achieves **zero-shot interpretable unlearning**. This is a key practical advantage over gradient-based methods, which require task-specific data and lack inherent interpretability, making DSG a uniquely transparent and data-efficient solution for mitigating hazardous knowledge, especially in data-scarce or zero-shot deployment.

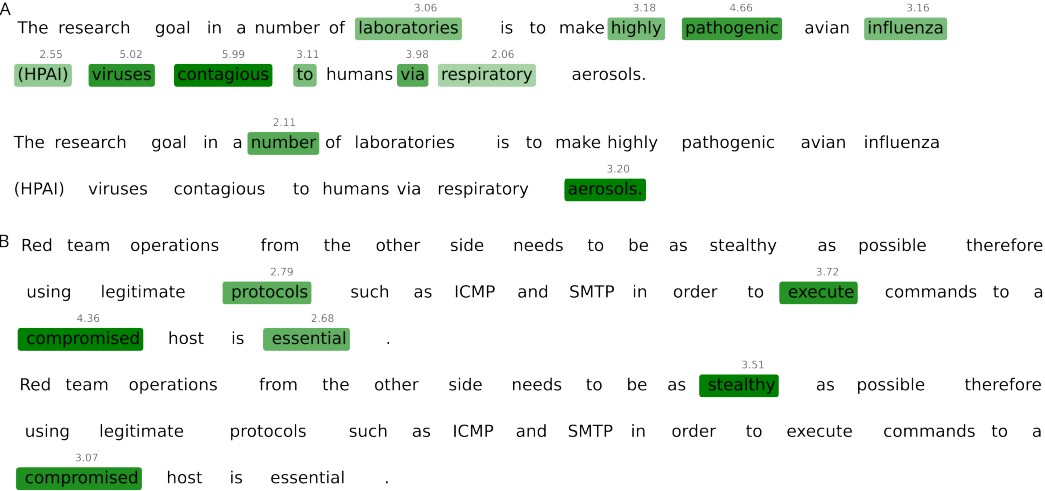

**Figure 13: Feature Activations on Example Sequences from Forget Sets.** (A) WMDP-Bio sequence with words highlighted in green indicating activation values > 0 for feature ID (top) 373 and (bottom) 10933. (B) WMDP-Cyber sequence with words highlighted in green indicating activation values > 0 for feature ID (top) 15286 and (bottom) 2905. Activation magnitudes are reported above the words in grey.

### Biology

| ID | Sentence |
|---|---|
| 12382 | Terms related to biological processes and structures in living organisms |
| 9722 | Concepts related to biological processes and systems |
| 343 | Terms related to biological processes and laboratory techniques |
| 373 | Scientific terminology related to biological processes and cellular functions |
| 11 | Scientific terms and concepts related to biology |
| 15969 | Terms related to biotechnology and bio-related fields |
| 12117 | Concepts related to biological or cellular processes and conditions, particularly focusing on requirements, limitations, and energy dynamics |
| 5877 | Terms related to biological processes and molecular interactions |
| 968 | Terms related to biological or medical processes and conditions, especially those involving cellular or molecular biology |
| 622 | Scientific terminology related to cellular processes and functions |
| 5231 | Specific terminology related to biological processes and gene expression |
| 10546 | Biological and genetic terms or sequences |
| 12037 | Medical terms and technical jargon related to genetic and biological research |
| 6150 | Elements related to scientific terminology, particularly in genetics and molecular biology |
| 5704 | Scientific terms and jargon related to biological research |
| 14747 | Technical terminology and references related to biotechnology and medical research |
| 8786 | Scientific terminology related to molecular biology and laboratory procedures |
| 10933 | Terms related to biological research and medical methodologies |
| 140 | Technical terms and concepts related to biology and bioengineering |
| 13527 | Terms related to biological or medical research, particularly focusing on specific conditions and associated microorganisms |

### Cybersecurity

| ID | Sentence |
|---|---|
| 15331 | Terms related to cyber threats and cybersecurity issues |
| 2060 | Explicit mentions of digital security concerns |
| 15286 | Concepts and terms related to digital security and data integrity |
| 11015 | Terms related to security and the act of securing something |
| 364 | References to security and related terms |
| 4836 | Concepts related to secure web connections and cryptocurrency surplus |
| 2905 | Terms related to data security and encryption |
| 10931 | References to national security and related governmental positions or actions |
| 11716 | Technical terms and language related to coding and software functionality, specifically focusing on vulnerabilities |
| 16160 | Discussions related to technology and computer systems |
| 6309 | References to technology and its applications across various sectors |
| 10543 | Keywords related to safety and security measures in various contexts |
| 11513 | Terms related to computing and data centers |
| 1803 | References to Common Weakness Enumeration (CWE) identifiers |
| 12681 | Keywords related to safety and security |
| 11520 | References to information technology and IT-related concepts |
| 11323 | Key concepts related to digital citizenship and its implications in various contexts |
| 10415 | Key components of data processing and communication in systems, particularly focusing on the details of data packet headers and their significance for routing and interpreting data |
| 3943 | References to computing systems and technologies |
| 4686 | References to technology and tech-related topics |

**Table 16: Top 20 SAE Features for Biology and Cybersecurity in Zero-Shot Setting.** List of the top 20 SAE feature IDs identified by querying Neuronpedia with "Biology" and "Cybersecurity", alongside their corresponding Neuronpedia-provided interpretations, showing the semantic relevance of the selected features to the targeted knowledge domains.

