# OpenReview forum: "SAEs Can Improve Unlearning: Dynamic Sparse Autoencoder Guardrails for Precision Unlearning in LLMs"
_colmweb.org/COLM/2025/Conference — COLM 2025_

### Official Review · Reviewer_mnFd · 2025-05-11

**Rating:** 7
**Confidence:** 5
**Ethics Flag:** 1

**Summary:**

### Quality

This paper presents a strong and technically sophisticated contribution to machine unlearning in large language models (LLMs). The authors propose **Dynamic SAE Guardrails (DSG)**, an activation-based unlearning method that combines Sparse Autoencoders (SAEs), Fisher Information–based feature selection, and a dynamic input-dependent classifier. Compared to gradient-based methods like RMU or NPO, DSG achieves superior trade-offs between *forgetting effectiveness* and *utility preservation*. The method is evaluated rigorously across multiple benchmarks (WMDP, MUSE), showing robustness to sequential unlearning, data scarcity, and relearning attacks. Detailed theoretical analysis, including Fisher Information approximations and causal interpretations of SAE features, strengthens the technical depth.

### Clarity

The paper is clearly written, using diagrams, step-by-step algorithms, and formal definitions to explain the core mechanisms of DSG. The proofs of key theoretical claims are relegated to the appendix but are referenced. Experimental setups are explained in detail, and ablation studies further clarify the contribution of each component of the method.

### Originality

The paper challenges prior conclusions that SAEs underperform for unlearning by showing that **context-sensitive dynamic interventions** can significantly improve outcomes. The novel combination of **Fisher Information as a proxy for causal influence** and **sequence-level activation thresholds** for conditional intervention constitutes a notable advance over prior static SAE methods. This is among the first works demonstrating effective zero-shot unlearning via interpretable SAE features.

### Significance

The method offers a practical and interpretable alternative to gradient-based unlearning, with advantages in computational efficiency, hyperparameter stability, and black-box applicability. It shows promise for real-world deployment scenarios requiring frequent or regulatory-mandated unlearning. By building on publicly available SAEs (e.g., Gemma-Scope), the work is positioned to be reproducible and extensible. The implications for safety, privacy, and responsible model control are substantial.

**Reasons To Accept:**

This paper presents a substantial and timely contribution to machine unlearning for large language models (LLMs). It introduces **Dynamic SAE Guardrails (DSG)**, a novel unlearning method that leverages the interpretability and sparsity of Sparse Autoencoders (SAEs) alongside a dynamic, context-sensitive classifier. The approach addresses long-standing challenges dominant gradient-based unlearning methods face, including high computational costs, hyperparameter instability, and poor performance in sequential or zero-shot unlearning scenarios.

DSG is theoretically well-grounded. The authors provide clear causal framing and justify their feature selection mechanism using Fisher Information as a proxy for causal influence. The dynamic thresholding mechanism minimizes side effects on general utility while enabling precise, input-dependent intervention.

The empirical results are strong and comprehensive. DSG consistently outperforms leading baselines such as RMU, NPO, SCRUB, and SimNPO on benchmarks like WMDP and MUSE. It demonstrates superior unlearning effectiveness while preserving model utility on tasks such as MMLU and MT-Bench. Furthermore, DSG is robust to sequential unlearning and resistant to relearning attacks, outperforming prior work even in adversarial finetuning scenarios.

Additional strengths include:

* Practical efficiency through reliance on forward passes only
* Strong performance in low-resource and zero-shot settings using interpretable features
* Extensive ablation studies that isolate the contribution of each component
* Reproducibility, with detailed algorithm descriptions and a commitment to code release

Overall, this paper makes an original and impactful contribution to COLM's goals of advancing safe, interpretable, and efficient language modeling. It sets a new direction for activation-based unlearning methods and expands the practical toolkit for maintaining and controlling deployed LLMs.

**Reasons To Reject:**

While the paper introduces an innovative and well-motivated approach to machine unlearning through Dynamic SAE Guardrails (DSG), several concerns may limit its readiness for publication at COLM:

1. **Limited generalization across model scales and architectures**
   All experiments are conducted on the `gemma-2-2b-it` model using Gemma-Scope SAEs. It remains unclear how well DSG transfers to other model families or scales. The current results, while strong, may not generalize to larger models or those with significantly different activation geometries or training regimes.

2. **Heavy reliance on pre-trained SAEs and interpretability assumptions**
   DSG assumes the availability and quality of well-trained Sparse Autoencoders. In practice, training high-quality SAEs is nontrivial, computationally expensive, and architecture-specific. The paper amortizes this cost by relying on open-source SAEs, but this limits general applicability. Furthermore, the method assumes that SAE features are semantically aligned and interpretable enough to support zero-shot unlearning—an assumption that may not hold in more complex domains.

3. **Overhead of dynamic intervention during inference**
   Although more efficient than backpropagation-based methods, DSG requires additional computation at inference time, including the computation of feature activations, classification via ρ(x), and conditional clamping. This inference-time overhead could be non-trivial in latency-sensitive applications, especially when applied across multiple layers or with larger feature dictionaries.

4. **Limited robustness analysis in adversarial settings**
   While DSG shows better resistance to relearning attacks in an API-based threat model, the robustness of its feature selection mechanism and classifier under white-box or obfuscation-based attacks has not been deeply investigated. The method’s reliance on fixed feature activations and thresholds could be vulnerable to adversarial inputs crafted to evade the dynamic guardrails.

5. **Theoretical assumptions not fully validated empirically**
   The use of Fisher Information as a proxy for causal influence is theoretically compelling, but its practical behavior is not explored in depth. For example, how often high FI features are causally responsible for generating the unwanted outputs is not empirically verified. The method may benefit from additional causal validation or alignment with feature attribution techniques.

6. **Limited qualitative analysis of interpretability**
   The paper promotes DSG as interpretable, especially in the zero-shot unlearning scenario. However, limited qualitative analysis or examples show how the selected SAE features correspond to meaningful concepts in practice. Including a few case studies or neuron activation visualizations could improve confidence in the interpretability claims.

---

> ### Author Response · Authors · 2025-05-31
>
> We thank Reviewer mnFd for their thorough and insightful review. We are particularly pleased that the reviewer recognized DSG as a novel unlearning method that effectively leverages the interpretability of SAEs. We appreciate their acknowledgment that our approach addresses key limitations of dominant gradient-based unlearning methods, including high computational costs, hyperparameter instability, and poor performance in sequential and zero-shot unlearning scenarios.
>
> We address the noted concerns below.
>
> **1. Limited generalization across model scales and architectures**
>
> We used Gemma-2-2B for these experiments for two primary reasons. Firstly, high-quality, publicly accessible SAEs for this model are available via SAE-Scope; these are widely used, have been thoroughly evaluated with well-vetted explanations (e.g., on Neuronpedia), offering a solid foundation for rigorously validating DSG's unlearning mechanism. Secondly, employing Gemma-2-2B ensures alignment with prior research [1] that also utilized this model for SAE-based unlearning, allowing us to show DSG's improvement more directly.
>
> Although our current experiments use Gemma-2-2B, DSG is not fundamentally limited by language model architecture or size. DSG operates on activations within the model's residual stream, and its core mechanisms, such as Fisher Information-based feature selection (derived from SAE reconstruction error) and the dynamic classifier, apply broadly across architectures. The SAE itself is a modular component. Adapting and scaling SAEs to larger language models follows well-established scaling laws (Fig. 4 in [2] shows larger LMs typically requiring wider SAEs for comparable reconstruction error). DSG can readily leverage advancements in both SAE architectures and training recipes (Table 1 in [5]) as they become available. Our primary focus here was to introduce and demonstrate the efficacy of this new unlearning *paradigm*—combining SAEs with principled feature selection and dynamic clamping—on an established and well-understood setup. We hope the community builds upon our findings and applies DSG to models of larger sizes and varied architectures. However, given DSG's novelty, its theoretical underpinnings, and the extensive empirical evidence convincingly demonstrating its superior forgetting-utility balance and robust performance over gradient-based baselines, we believe this work offers a significant and timely contribution to the field of machine unlearning.
>
> Finally, we believe our work resonates with the COLM guidelines on computational resources. These guidelines value research for its potential impact, even if initial studies like ours (demonstrating a novel, effective unlearning method) use accessible model scales rather than larger models that require extensive resources.
>
> **2. Heavy reliance on pretrained SAEs and interpretability assumptions**
>
> We acknowledge that our work currently leverages publicly available, high-quality SAEs—a practical approach similar to the broader field's reliance on large, pretrained foundation models. We anticipate a growing ecosystem where robust SAEs for various foundation models are increasingly trained and shared. SAEs exhibit characteristics conducive to modularity and adaptation: research [3] demonstrates their transferability between base and chat models, and work like [4] shows they can be finetuned to capture specialized information in specific subdomains, much like pretrained language models. A core contribution of our work is to demonstrate that these SAEs, when appropriately leveraged with a mechanism like DSG, serve as a highly effective foundation for practical and interpretable unlearning.
>
> The reviewer raises a concern that SAE features may not be semantically aligned and interpretable enough to support zero-shot unlearning in complex domains. In this work, we show that SAE features capture interpretable information in real-world domains such as those covered by our news, books, biology, and cybersecurity datasets. Furthermore, recent research [9] demonstrates SAEs' capability to capture relatively sophisticated concepts like refusal, multilinguality, and arithmetic. The aspirational goal of SAEs, as explored in [7], is to comprehensively represent all of an LM's internal features with sufficiently wide SAEs. While achieving universal, perfect SAE interpretability remains an active research frontier, the field is making rapid progress. This progress includes: improved SAE training methodologies and architectural developments [5]; enhanced capture of rarer or more nuanced features [4]; and more sophisticated automated interpretability techniques [6]. Our approach to zero-shot unlearning directly benefits from these advances, enabling targeted interventions based on semantic understanding.

---

> > ### Author Response · Authors · 2025-05-31
> > **Official Comment by Authors (Continued)**
> >
> > **3. Overhead of dynamic intervention during inference**
> > While DSG introduces additional computations, including SAE feature activations,  $\rho(x)$  calculation, and conditional clamping, our empirical results in **Appendix L and Table 15** show minimal performance impact. Applying DSG with a Gemma-Scope SAE (16k feature dictionary) to layer 3 of Gemma-2-2B-it increases latency by only 7.3% for 256 tokens and 3.6% for 512/1024 tokens on an A6000 GPU. Our experiments also show that effective unlearning is achieved by applying DSG to just a single, early model layer and extensive multi-layer intervention is unnecessary for strong performance.  The bulk of the observed overhead stems from the SAE's matrix multiplications, which are amenable to standard inference optimizations including quantization and specialized kernels. DSG's dynamic classification logic itself adds negligible latency beyond the standard SAE forward pass, making the primary computational bottleneck readily addressable through existing acceleration techniques.
> >
> > Gradient-based methods avoid per-inference overhead once unlearning is complete but incur substantial finetuning costs for each unlearning request. DSG offers a compelling trade-off through minimal, optimizable inference overhead in exchange for significantly more efficient unlearning operations. This advantage becomes particularly pronounced in production environments requiring frequent content updates, where the cumulative costs of repeated gradient-based retraining would be prohibitive.
> >
> >
> > **4. Limited robustness analysis in adversarial settings**
> >
> >
> > DSG is designed for API-based, black-box threat models where attackers lack visibility into internal architecture, including selected features ($S_{n_{feats}}$), dynamic threshold ($\tau$), or $\rho(x)$ calculation methodology. This operational opacity creates substantial barriers for adversaries attempting to craft inputs that simultaneously elicit undesired behavior and manipulate $\rho(x)$ to bypass guardrails. While the reviewer characterizes DSG as relying on fixed feature activations and thresholds, this overlooks the practical challenges facing black-box attackers who must first discover specific features, threshold values, and calculation procedures through API interactions. The dynamic nature of $\rho(x)$ calculation across sequences further complicates attacks by requiring coordinated manipulations across multiple tokens while maintaining semantic coherence.
> >
> > As acknowledged in our paper, works such as [8] demonstrate that latent-space defenses can be circumvented, but their attack methodologies presume substantially greater access than DSG's defined threat model provides, including white-box model gradients or gray-box access to monitor internal probe (DSG) outputs. Such access levels exceed the API-only scenario that DSG addresses. In an API-only deployment, the system does not provide the internal feedback mechanisms that these attacks require. Investigating DSG's robustness under stronger threat models represents important future research, distinct from the API-only deployment scenario where DSG demonstrates effective resistance within its intended operational parameters.

---

> > > ### Author Response · Authors · 2025-05-31
> > > **Official Comment by Authors (Continued)**
> > >
> > > **5. Theoretical assumptions not fully validated empirically (FI as Causal Proxy)**
> > >
> > > We appreciate the reviewer highlighting the importance of empirically validating the link between our Fisher Information (FI) and causal influence. While our theorems provide a principled foundation, DSG's existing empirical results offer evidence for FI's practical effectiveness through multiple lines of convergent evidence:
> > >
> > > * **Superiority of FI-based Selection (Ablation, Figure 7A):** As shown in **Figure 7A**, holding the number of selected features constant, DSG's FI-based selection achieves significant additional reduction (about 8\% on average) in WMDP-Bio forget accuracy compared to sparsity-only selection (as per [1]), with identical MMLU utility preservation. This superior performance strongly suggests that FI better identifies causally instrumental features responsible for generating unwanted outputs.
> > >
> > > * **Performance Scaling with FI-Selected Feature Count (Ablations, Figure 7C & Appendix K):** The systematic impact of varying $n_{feats}$ provides additional validation. An optimal range for $n_{feats}=20$ maximizes unlearning without excessive utility loss, indicating that FI ranking captures meaningful importance notions rather than arbitrary correlations.
> > >
> > > * **Alignment of Semantically-Selected Zero-Shot Features with FI-Ranked Features:** DSG's success in zero-shot unlearning using semantically selected features also provides compelling validation. We observe notable overlap (>87\%) between these interpretable, domain-relevant features and those highly ranked by our FI-based importance score. We will add quantitative details of this overlap to the revised manuscript. Since semantic selection naturally correlates with causal importance for domain-specific knowledge, this alignment strongly supports that high FI features are indeed causally responsible for generating targeted outputs.
> > >
> > > The convergent evidence from direct performance comparisons, scaling behavior, and semantic alignment collectively strengthens confidence in FI's effectiveness as a causal influence proxy within our experimental framework. We leave more granular causal intervention studies and integration with established feature attribution techniques as valuable directions for future work.
> > >
> > >
> > >
> > > **6. Limited qualitative analysis of interpretability**
> > >
> > >
> > > We appreciate the reviewer highlighting this important aspect of our interpretability claims. **Appendix M ("Feature Interpretability"), along with Figure 13 and Table 16,** provides qualitative evidence addressing how selected SAE features correspond to meaningful concepts in practice.
> > >
> > > **Table 16 ("Top 20 SAE Features for Biology and Cybersecurity in Zero-Shot Setting")** presents the selected zero-shot feature IDs obtained through Neuronpedia queries alongside their corresponding semantic interpretations. We find that these features show clear alignment with domain-relevant concepts, with biology features present that correspond to "biological processes and cellular functions," "genetic and molecular biology," and "biotechnology research," and cybersecurity features present that align with "cyber threats and digital security," "data integrity and encryption," and "information security measures." This establishes that semantically selected features capture relevant conceptual categories.
> > >
> > > **Figure 13 ("Feature Activations on Example Sequences from Forget Sets")** shows feature activation visualizations and the correspondence between feature activations and meaningful concepts in real-world examples. The visualizations show that biology-relevant features consistently activate on hazardous domain-specific terms such as "pathogenic," "contagious," "viruses," and "influenza" within WMDP-Bio sequences, while cybersecurity features respond predictably to relevant terminology including "protocols," "compromised," and "stealth" in WMDP-Cyber content. These activation patterns confirm that selected features respond meaningfully to conceptually relevant content rather than spurious correlations.
> > >
> > > The qualitative evidence presented in **Appendix M** establishes that SAE features leveraged by DSG correspond to interpretable concepts. The success of zero-shot unlearning using these semantically meaningful features provides additional validation that the selected features capture genuine conceptual relationships within the targeted knowledge domains.

---

> > > > ### Author Response · Authors · 2025-05-31
> > > > **Official Comment by Authors (Continued)**
> > > >
> > > > ---
> > > > **References**
> > > >
> > > > [1] Applying sparse autoencoders to unlearn knowledge in language models (Farrell et al. 2024).
> > > > [2] Scaling and evaluating sparse autoencoders (Gao et al. 2025).
> > > > [3] SAEs (usually) Transfer Between Base and Chat Models (Kissane et al. 2024).
> > > > [4] Decoding Dark Matter: Specialized Sparse Autoencoders for Interpreting Rare Concepts in Foundation Models (Muhamed et al. 2024).
> > > > [5] A Survey on Sparse Autoencoders: Interpreting the Internal Mechanisms of Large Language Models (Shu et al. 2025).
> > > > [6] Enhancing Automated Interpretability with Output-Centric Feature Descriptions (Gur-Arieh et al. 2025).
> > > > [7] Scaling Monosemanticity: Extracting Interpretable Features from Claude 3 Sonnet (Templeton et al. 2024).
> > > > [8] Obfuscated Activations Bypass LLM Latent-Space Defenses (Bailey et al. 2024).
> > > > [9] On the Biology of a Large Language Model (Lindsey et al 2025).

---

> > ### Comment · Reviewer_mnFd · 2025-06-04
> >
> > Thanks to the author for their feedback. My questions and doubts have been addressed, and I am looking forward to seeing the better version of the paper with the addition mentioned, included

---

> > > ### Author Response · Authors · 2025-06-05
> > >
> > > We thank the reviewer for their positive follow-up comment. We are pleased to hear that our responses have successfully addressed their questions and doubts. We will ensure that all discussed additions are carefully incorporated into the final manuscript.

---

### Official Review · Reviewer_teuD · 2025-05-12

**Rating:** 6
**Confidence:** 3
**Ethics Flag:** 1

**Summary:**

The paper studies unlearning with sparse autoencoders. The authors propose a dynamic sparse autoencoder that performs sparse feature selection with Fisher information and dynamically determines thresholds for activation clamping. It provides both theoretical justifications and extensive empirical evaluations.

**Reasons To Accept:**

DSG performs well on recent unlearning standard benchmarks.

**Reasons To Reject:**

- The paper is a little bit of dense for me. Some parts are unclear (e.g., digits on the green/blue dash lines in Figure 3).
- The experiments set D_{forget} and D_{retain} as texts from different domains. Maybe somewhere in the appendix, we discuss whether the unlearning algorithms perform if they are from the same domain (e.g., removing a subset of WMDP-Bio and checking the performances on the remaining subset)
- Some previous works also investigate Fisher information in the unlearning ([1], [2]).

[1] Eternal Sunshine of the Spotless Net: Selective Forgetting in Deep Networks, CVPR 2020
[2] Unlearning with Fisher Masking, arxiv 2023

---

> ### Author Response · Authors · 2025-05-30
>
> We thank the reviewer for their constructive feedback and for recognizing the strengths of DSG against gradient-based baselines on standard unlearning benchmarks. We address the points raised below.
>
> **1. Regarding Paper Clarity and Figure 3 Annotations**
>
> We appreciate the reviewer's feedback on the paper's density. We aimed to provide a comprehensive account of both theoretical justifications and empirical results, and we will incorporate more intuitive explanations and key takeaways in the final version to enhance overall clarity.
>
> Specifically for **Figure 3**, the digits on the green and blue dashed lines represent the **clamp strengths ($c$)** used for those particular experimental configurations. This is also noted in the figure caption ("Clamp strengths (c) used for DSG points are shown as annotations"). These annotations help illustrate how varying the clamp strength impacts the trade-off between unlearning efficacy and MMLU accuracy.
>
> **2. Regarding Domain Specificity ($D_{\text{forget}}$ and $D_{\text{retain}}$)**
>
> The reviewer raised a concern about experiments where $D_{\text{forget}}$ and $D_{\text{retain}}$ are from different domains and suggested discussing performance when they are from the same domain. We would like to clarify that DSG has indeed been evaluated in such settings:
>
> * **MUSE Experiments Showcase Same-Domain Performance:** While our WMDP experiments use data from different domains, our MUSE experiments (Section 4.2) specifically address scenarios with data from the same or very similar domains.
>     * For the **NEWS corpus**, both $D_{\text{forget}}$ and $D_{\text{retain}}$ are drawn as disjoint random subsets from the *same distribution* of BBC news articles.
>     * For the **BOOKS corpus**, $D_{\text{forget}}$ (Harry Potter books) and $D_{\text{retain}}$ (Harry Potter FanWiki content) originate from a highly similar knowledge domain.
>     DSG's strong performance in these same-domain/close-domain MUSE settings (detailed in **Table 2 and Section 4.2**) underscores its applicability beyond cases with stark domain differences. We will ensure this is highlighted more explicitly in the revised manuscript.
>
> * **General Prerequisite for Unlearning:** We would also like to emphasize that *all* approximate unlearning methods inherently depend on the model's ability to distinguish, at some level, between forget and retain data. There must exist a model-computed statistic $s(x)$ (for gradient methods, this is $s_{\text{grad}}(x) = \nabla_{\theta}\ell(M_{\theta};x)$; for DSG, $s_{\text{act}}(x)$ is derived from SAE activations leading to $\rho(x)$) whose distributions differ for $D_{\text{forget}}$ and $D_{\text{retain}}$ (i.e., $TV(P(s|D_{\text{forget}}), P(s|D_{\text{retain}})) > 0$). Gradient-based unlearning baselines such as RMU [1], which are trained to inject noise when forget data is detected, are *effective* only if such a separation provides a discernible signal to differentiate between forget and retain data; without it, their selective actions would be arbitrary.
>
> * **DSG's Transparency and Controllability:** A key advantage of DSG is that it makes this separability assumption explicit and controllable. This is achieved via the Neyman-Pearson optimal test on $\rho(x)$ with a user-tunable threshold $\tau$, directly controlling the False Positive Rate (FPR). This contrasts with gradient-based methods, where the classifier distinguishing forget/retain data is implicitly embedded within the model weights.
>
> **3. Regarding Prior Work on Fisher Information**
>
> We thank the reviewer for referencing [2] and [3], which investigate Fisher Information (FI) for unlearning and we will cite these in our revision. DSG's application of FI is distinct due to its nature as an **activation-based unlearning method**:
>
> * Both  [2] (e.g., "FisherNoise," modifying model weights) and [3] (masking LLM weights based on FI contributions) use FI to guide direct modifications or masking of the **LLM's weights**.
> * In contrast, DSG uniquely applies an FI-inspired metric (based on SAE reconstruction error) to score and select interpretable **SAE features**. Our intervention is then the dynamic clamping of these selected features' **activations**, not direct manipulation or masking of LLM weights.
> * This positions DSG as a novel approach that targets specific, interpretable features within a modular SAE for intervention at the activation level. Our SSD baseline (cited in our paper), which more closely resembles the direction of directly modifying weights based on FI, is outperformed by DSG, highlighting the benefits of our targeted activation-clamping strategy.
>
>
> ---
> **References**
>
> [1] The WMDP Benchmark: Measuring and Reducing Malicious Use With Unlearning (Li et al. 2024).
> [2] Eternal Sunshine of the Spotless Net: Selective Forgetting in Deep Networks (Golatkar et al. 2020).
> [3] Unlearning with Fisher Masking (Liu et al. 2023).

---

> > ### Author Response · Authors · 2025-06-09
> > **Summary**
> >
> > Dear Reviewer teuD,
> >
> > Thank you again for your constructive feedback. We are following up on our rebuttal, which addresses the main points you raised.
> >
> > To summarize our key clarifications:
> >
> > * **On Domain Specificity:** Regarding your concern about testing on same-domain data, we clarified that our MUSE experiments were designed for this exact scenario. For instance, the NEWS corpus uses forget/retain sets drawn as disjoint subsets from the same distribution of BBC news articles, and the BOOKS corpus uses content from a highly similar knowledge domain. DSG's strong performance in these settings underscores its applicability beyond cases with stark domain differences.
> >
> > * **On Prior Work with Fisher Information:** We clarified that DSG's application of Fisher Information is distinct from this prior work. While those methods use FI to guide direct modifications or masking of the LLM's **weights**, our approach uniquely uses an FI-inspired metric to select interpretable SAE features for intervention at the **activation** level. The benefit of this targeted activation-clamping strategy is highlighted by DSG outperforming our SSD baseline, a method cited in our paper that more closely resembles the FI-based weight-modification approach.
> >
> > * **On Figure 3:** To answer your specific question about Figure 3, we clarified that the digits on the lines represent the clamp strengths (`c`).
> >
> > We hope this summary is helpful, and we are happy to discuss any further questions you may have.

---

### Official Review · Reviewer_5A4g · 2025-05-14

**Rating:** 7
**Confidence:** 2
**Ethics Flag:** 1

**Summary:**

This paper introduce Dynamic Sparse Autoencoder Guadrails(DSG) for precision unlearning by leavaging principled feature selection with fisher information-based feature importance scoring and a dynamic classifier which are input-dependent.  The experimental results shows that the proposed DSG could substantially outperforms leading unlearning methods, achieving superior forgetting-utility balance.

**Questions To Authors:**

** Reviewer Note: ** after the rebuttal, I have improved the score from 6 to 7.

**Reasons To Accept:**

1. The proposed methods of combining activation based SAE with feature selection and dynamic classifier is reasonable and helpful, which outperform gradient-based methods across multoiple tasks.
2. The experiments and analysis are extensive and thought-provoking.

**Reasons To Reject:**

Comparing to NPO and RMU(running on 7b model), the experiments is based on relative smaller model (gemma-2-2b), it is unclear how the performance on larger models.

---

> ### Author Response · Authors · 2025-05-30
>
> We thank the reviewer for recognizing the strengths of DSG, including extensive, thought-provoking experiments, and for noting DSG's strong performance against gradient-based methods.
>
> We used Gemma-2-2B for these experiments for two primary reasons. Firstly, high-quality, publicly accessible SAEs for this model are available via SAE-Scope; these are widely used, have been thoroughly evaluated, and provide well-vetted explanations (e.g., on Neuronpedia), offering a solid foundation for rigorously validating DSG's unlearning mechanism. Secondly, employing Gemma-2-2B ensures alignment with prior research [1] that also utilized this model for SAE-based unlearning, allowing us to show DSG's improvement more directly.
>
> Although our current experiments use Gemma-2-2B, DSG is not fundamentally limited by language model architecture or size. DSG operates on activations within the model's residual stream, and its core mechanisms, such as Fisher Information-based feature selection (derived from SAE reconstruction error) and the dynamic classifier, apply broadly across architectures. The SAE itself is a modular component. Adapting and scaling SAEs to larger language models follows well-established scaling laws (Fig. 4 in [2] shows larger LMs typically requiring wider SAEs for comparable reconstruction error). DSG can readily leverage advancements in both SAE architectures and training recipes as they become available (Table 1 in [3]). Our primary focus here was to introduce and demonstrate the efficacy of this new unlearning *paradigm*—combining SAEs with principled feature selection and dynamic clamping—on an established and well-understood setup. We hope the community builds upon our findings and applies DSG to models of larger sizes and varied architectures. However, given DSG's novelty, its robust theoretical underpinnings, and the extensive empirical evidence convincingly demonstrating its superior forgetting-utility balance and robust performance over gradient-based baselines, we believe this work offers a significant and timely contribution to the field of machine unlearning.
>
> Finally, we believe our work resonates with the COLM guidelines on computational resources. These guidelines value research for its potential impact, even if initial studies like ours (demonstrating a novel, effective unlearning method) use accessible model scales rather than larger models that require extensive resources.
>
> ---
> **References**
>
> [1] Applying sparse autoencoders to unlearn knowledge in language models (Farrell et al 2024).
> [2] Scaling and evaluating sparse autoencoders (Gao et al 2025).
> [3] A Survey on Sparse Autoencoders: Interpreting the Internal Mechanisms of Large Language Models (Shu et al 2025).

---

> > ### Comment · Reviewer_5A4g · 2025-06-05
> >
> > Thanks for the authors' response. I agree the smaller model still show case the potential, a decent discuss for this limitations in the future version will make it better.

---

> > > ### Author Response · Authors · 2025-06-05
> > >
> > > We sincerely thank the reviewer for their constructive follow-up, thoughtful reassessment, and updated evaluation of our work. We are encouraged that the reviewer agrees our current experiments effectively demonstrate the method's potential with the model scale used.
> > >
> > > Following the reviewer's valuable suggestion, the revised manuscript will include a dedicated discussion on the current model scale, its implications, and avenues for future research within the limitations section, to further strengthen the paper.

---

### Official Review · Reviewer_JnnW · 2025-05-20

**Rating:** 7
**Confidence:** 4
**Ethics Flag:** 1

**Summary:**

The paper presents Dynamic SAE Guardrails (DSG), a novel approach to LLM unlearning that leverages Fisher Information-based feature selection within an intermediate SAE and applies targeted feature clamping using a dynamic, input-dependent classifier. The method is theoretically motivated by two key insights: (1) the relationship between Fisher Information and Causal Feature Importance, and (2) the optimality of threshold-based feature selection. Experimental results on the WMDP and MUSE benchmarks highlight DSG's superior unlearning performance, along with notable advantages such as robustness to hyperparameters, data efficiency, and strong resistance to relearning attacks.

**Questions To Authors:**

- [Q1] I believe this is based on previous work [C], but could you elaborate on the choice to ablate features with $f_j = -c$ instead of $f_j = 0$? Intuitively, since JumpReLU suppresses activations below a threshold to zero, a negative activation would still activate the unwanted feature but in the opposite direction. How does this approach avoid the risk of over-unlearning?
- [Q2] In Section 3.3, is there a specific reason for choosing sequence-level interventions over token-level ones? Token-level interventions might offer more fine-grained control and potentially reduce degradation in model utility. Was this trade-off considered?
- [Q3] The presentation of Theorem 3 seems somewhat abrupt. Could it be moved before the Conditional Clamping paragraph with some text describing its implications?
- Line 79: incorrect font for $v_i \in \mathbb{R}^{d_{model}}$

[C] Applying sparse autoencoders to unlearn knowledge in language models. Safe Generative AI Workshop @ NeurIPS 2024.

**Reasons To Accept:**

- [S1] **Strong empirical performance.** DSG demonstrates clear improvements over existing unlearning methods, not only in overall unlearning efficacy but also in challenging scenarios such as resistance to relearning attacks and sequence-level unlearning. The experimental results are thorough and provide valuable insights into the method’s strengths.
- [S2] **Interesting theoretical grounding.** The use of causal intervention concepts and statistical optimality to guide the design of the unlearning framework is both innovative and compelling. These theoretical contributions are likely to be of significant interest to the unlearning research community.

**Reasons To Reject:**

- [W1] **Applicability limited to black-box adversaries.** A core concern is that DSG primarily defends against black-box adversaries with no access to model parameters. This limits its alignment with the broader goal of unlearning, which is to produce a model indistinguishable from one trained solely on retained data. Since DSG relies on a filtering mechanism external to the model weights, a white-box adversary could potentially bypass the guardrails by disabling the filter, thus undermining the unlearning guarantees.
- [W2] **Potential domain specificity.** The effectiveness of DSG’s thresholding mechanism appears to rely on a clear separation in token activation distributions between the forget and retain sets, as seen in the WMDP vs. WikiText case (Figure 2). However, it’s unclear whether this separation persists in more realistic or fine-grained scenarios, such as when the two sets are from closely related domains (e.g., TOFU [A]). Additional analysis would help determine whether DSG remains effective when this activation distinction is much less pronounced.
- [W3] **Concerns about MUSE experimental results.** The target model scores reported in Table 2 are substantially lower than those in the original MUSE paper (e.g., 21.15 in Table 2 vs. 58.4 in Table 3 of [B] on NEWS C1). While this may be due to differences in the base LLM, such a large discrepancy raises questions about whether DSG’s strong unlearning results are due to the method itself or the use of an underperforming model. Additionally, the experiments omit a retrained baseline, which is typically used as an oracle to contextualize unlearning effectiveness.

[A] TOFU: A Task of Fictitious Unlearning for LLMs. COLM 2024.\
[B] MUSE: Machine Unlearning Six-Way Evaluation for Language Models. ICLR 2025.

---

> ### Author Response · Authors · 2025-05-30
>
> We thank the reviewer for recognizing DSG's strong empirical performance and theoretical grounding.
>
> **[W1] Applicability limited to black-box adversaries:**
> DSG is designed to offer a practical and effective unlearning solution for the predominant LLM deployment scenario of API-based access. In this scenario, users and potential adversaries interact with the model in a black-box or gray-box (e.g., API-based fine-tuning) manner, without direct access to model weights [1]. Our experiments confirm DSG's robust unlearning and resilience within this common threat model.
>
> While the original motivation of unlearning has been to produce a model indistinguishable from one trained solely on $D_{\text{retain}}$,  it is typically infeasible to attain such a baseline in practice for LLMs due to the expense of (re)-training and the difficulty of completely isolating the forget/retain sets. Thus, existing approximate LLM unlearning research and associated benchmarks almost universally focus on evaluating methods based on their efficacy in achieving output-level forgetting of $D_{\text{forget}}$ while preserving utility on $D_{\text{retain}}$. DSG demonstrably outperforms existing baselines against these practical and widely adopted output-level metrics. We also note that there are other LLM unlearning works [4,8] that adopt non-finetuning based approaches but underperform gradient-based baselines.
>
> *White-box setting*: The reviewer is correct that in the completely white-box setting, it may be possible to remove DSG's filter. However, recent work has shown that it is also quite easy to perturb gradient-based unlearned models through simple techniques such as quantization [10] or few-shot finetuning on adversarial data [11] to attain pre-unlearning performance, and so it is not clear that alternatives yet provide a meaningful advantage in this challenging setting. We will clarify this in the draft and add relevant discussion to the revised manuscript.
> We also note that DSG can be conceptualized as a dynamic internal guardrail that operates on SAE feature activations, rather than solely on input/output filtering like more traditional guardrails [2]. In light of this, an interesting approach in the white-box setting may be to consider that any dynamically intervened activation state A' produced by DSG can, in principle, be realized by an equivalent model weight configuration W'. Distilling such an intrinsic W' from DSG's interventions is a promising area of future research.
>
> **[W2] Potential domain specificity:**
> We would like to note that DSG has been evaluated in settings where $D_{\text{forget}}$ and $D_{\text{retain}}$ are closely related.
> * **MUSE Experiments:** Although our WMDP experiments utilize $D_{\text{forget}}$ and $D_{\text{retain}}$ from different domains, our MUSE experiments (Section 4.2) specifically address scenarios involving data from the same or very similar domains. For instance, in the NEWS corpus, both $D_{\text{forget}}$ and $D_{\text{retain}}$ are drawn as disjoint random subsets from the same distribution of BBC news articles. For the BOOKS corpus, $D_{\text{forget}}$ (Harry Potter books) and $D_{\text{retain}}$ (Harry Potter FanWiki content) originate from a similar knowledge domain. DSG's strong performance in these same-domain/close-domain MUSE settings underscores its applicability beyond cases with stark domain differences. We will clarify this further in the revised manuscript.
> * **General Prerequisite for Unlearning:** We would also like to emphasize that *all* approximate unlearning methods inherently depend on the model's ability to distinguish, at some level, between forget and retain data. There must exist a model-computed statistic $s(x)$ (for gradient methods, this is $s_{\text{grad}}(x) = \nabla_{\theta}\ell(M_{\theta};x)$; for DSG, $s_{\text{act}}(x)$ is derived from SAE activations leading to $\rho(x)$) whose distributions differ for $D_{\text{forget}}$ and $D_{\text{retain}}$ (i.e.,$TV(P(s|D_{\text{forget}}), P(s|D_{\text{retain}})) > 0$). Gradient-based unlearning baselines such as RMU [3], which are trained to inject noise when forget data is detected, are *effective* only if such a separation provides a discernible signal to differentiate between forget and retain data.
> * **DSG's Transparency and Controllability:** A key advantage of DSG is that it renders this separability assumption explicit and controllable. This is achieved via the Neyman-Pearson optimal test on $\rho(x)$ with a user-tunable threshold $\tau$, which directly controls the False Positive Rate (FPR). This approach contrasts with gradient-based methods, where the classifier distinguishing forget/retain data is implicitly embedded within the model weights.

---

> > ### Author Response · Authors · 2025-05-31
> > **Official Comment by Authors (Continued)**
> >
> > **[W3] Concerns about MUSE experimental results:**
> > * **Base Model and Fine-tuning:** The lower absolute scores for our target model in MUSE (Table 2), when compared to the Llama-2 7B model in the original MUSE paper [6], are attributable to our use of a smaller Gemma-2-2B base model and a finetuning phase of only 5 epochs.
> > * **Relative Performance:** It is important to note that all baselines in our MUSE experiments were run on this *identical* Gemma-2-2B target model. DSG's strong *relative* unlearning performance over these baselines is therefore a direct indicator of its efficacy. A lower-performing target model offers a smaller margin for improvement, making DSG's observed lead more notable.
> > * **Retrained Baseline:** Including a fully retrained *oracle* baseline is often computationally prohibitive for LLM unlearning and is not a universal standard (e.g., the WMDP benchmark [3] does not mandate it). For the MUSE experiments, creating such a baseline was further complicated by the lack of publicly available SAEs for the original Llama-2-7B MUSE target model. Additionally, our Gemma-2-2B model (with a 2023 data cutoff) likely encountered NEWS/BOOKS content during its pretraining. This makes the creation of a *from-scratch* $D_{\text{retain}}$-only model non-trivial; indeed, the MUSE forget metrics of the base pretrained Gemma-2-2B model were only slightly lower than our fine-tuned target model (before any unlearning of $D_{\text{forget}}$).
> >
> >
> > **[Q1] Choice to ablate features with $f_j = -c$ instead of $f_j = 0$:**
> > This choice is rooted in prior empirical observations [7], where clamping SAE features to $0$ was found to result in no noticeable modification to the model. Specifically, they observed that with JumpReLU SAEs, clamping to $0$ still yielded a >0.99 probability for the original answer, whereas negative values (-10 to -15) began to effectively shift these probabilities. This suggests that to counteract a feature's influence, a negative value may be necessary to actively cancel out other interacting feature vectors in this direction. Furthermore, current unlearning benchmarks require models to achieve very low accuracy on forget sets, a target that higher (i.e., more negative) clamp strengths empirically help attain.
> >
> > Over-unlearning is hard to measure in practice, and thus, approximate unlearning typically focuses on improving the forget/utility trade-off. Controlling this remains a challenge inherent to all unlearning methods. For instance, gradient-based approaches often rely on heuristics like early stopping, while DSG offers a more transparent mechanism to manage the forget/utility trade-off through explicit control over the dynamic threshold $\tau$.
> >
> > **[Q2] Sequence-level vs. token-level interventions:**
> > We did consider and experimentally compare token-level interventions against our sequence-level approach. On current benchmarks (MUSE, WMDP), while token-level clamping offered more granular control, it also introduced a clamp strength $c$ dependency that appeared dataset-dependent. This additional complexity did not translate into a clear corresponding benefit in final forget/utility metrics for the benchmarks we considered in this work. However, we acknowledge that for future benchmarks measuring additional aspects, such as fluency during forget-domain related tasks, token-level interventions could prove more beneficial.
> >
> > **[Q3] Presentation of Theorem 3:**
> > Thank you for this suggestion. We agree that moving Theorem 3 (Neyman-Pearson Optimality) to appear before the Conditional Clamping paragraph in Section 3.3, accompanied by a brief explanation of its implications, would improve the paper's clarity. We will make this revision.
> >
> > **Line 79: incorrect font**:
> > Thank you; we will correct this typo in the revision.
> >
> >
> > **References**
> >
> > [1] Unveiling the Landscape of LLM Deployment in the Wild: An Empirical Study (Hou et al. 2025).
> > [2] Building Guardrails for Large Language Models (Dong et al. 2024).
> > [3] The WMDP Benchmark: Measuring and Reducing Malicious Use With Unlearning (Li et al. 2024).
> > [4] Guardrail Baselines for Unlearning in LLMs (Thaker et al. 2024).
> > [5] TOFU: A Task of Fictitious Unlearning for LLMs (Maini et al. 2024).
> > [6] MUSE: Machine Unlearning Six-Way Evaluation for Language Models (Shi et al. 2025).
> > [7] Applying sparse autoencoders to unlearn knowledge in language models (Farrell et al. 2024).
> > [8] In-Context Unlearning: Language Models as Few-Shot Unlearners (Pawelczyk et al 2024).
> > [9] Unlearning or Obfuscating? Jogging the Memory of Unlearned LLMs via Benign Relearning (Hu et al 2025).
> > [10] Catastrophic Failure of LLM Unlearning via Quantization (Zhang et al 2025).
> > [11] An Adversarial Perspective on Machine Unlearning for AI Safety (Lucki et al 2024).

---

> > > ### Comment · Reviewer_JnnW · 2025-06-11
> > >
> > > Thank you to the authors for your detailed responses. Many of my concerns have been addressed. While I still have some concerns regarding the white-box unlearning setup, I agree that this constitutes a broader challenge that can be explored in future work. Overall, I find the current contribution sufficient for publication and have updated my score to a 7.

---

> ### Author Response · Authors · 2025-06-09
> **Summary**
>
> Dear Reviewer JnnW,
>
> Thank you again for your detailed feedback. We are following up on our rebuttal, which addresses the core concerns you raised.
>
> To summarize our key clarifications:
>
> * **On Applicability (W1):** In addressing the concern about the *broader goal of unlearning*, we clarified that DSG is intentionally designed for the API-based deployment scenario (as discussed in the paper, lines 653-654) precisely because the ideal of creating an indistinguishable model is often infeasible due to prohibitive retraining costs of LLMs. The field has therefore adopted more practical, output-level benchmarks for unlearning, and on these benchmarks and metrics, DSG demonstrably outperforms other methods. Regarding the theoretical white-box setting, we noted this is a broader challenge where no current approximate unlearning method has a clear advantage, given that even gradient-based unlearned models can be easily perturbed to restore pre-unlearning performance.
>
> * **On Domain Specificity (W2):** Our MUSE experiments were designed to evaluate the fine-grained scenarios you described. For example, the forget/retain sets in the NEWS corpus are drawn from the same distribution of BBC news articles, and the BOOKS corpus uses content from a highly similar knowledge domain. DSG shows strong performance in these settings.
>
> * **On MUSE Experimental Results (W3):** The lower absolute scores are attributable to our use of a smaller Gemma-2-2B base model and a finetuning phase of only 5 epochs. Since all baselines were run on this identical target model, DSG's strong *relative unlearning performance* over these baselines is a direct indicator of its efficacy.
>
> We hope our response addresses your concerns, and we are happy to discuss any further questions you may have.

---

### Author Response · Authors · 2025-05-30

We sincerely thank all the reviewers for their thoughtful and detailed feedback on our work. We are encouraged that the reviewers recognized the strong empirical performance of Dynamic SAE Guardrails (DSG), its theoretical grounding, and its potential as a substantial and timely contribution to machine unlearning for LLMs. We particularly appreciate the acknowledgement of DSG's advantages in addressing key limitations of prevailing gradient-based methods.




Our primary motivation with DSG was to introduce a novel activation-based unlearning paradigm that leverages the interpretability of SAEs to achieve precise, efficient, and robust unlearning. We are pleased the reviewers found DSG's ability to consistently outperform leading baselines on challenging benchmarks like WMDP and MUSE, while demonstrating superior utility preservation, to be a key strength. Furthermore, DSG's distinct advantages such as:

*   **Practical Efficiency:** Relying only on forward passes for intervention, avoiding costly gradient computations inherent in many unlearning techniques.
*   **Enhanced Stability and Robustness:** Showing greater hyperparameter stability and robust performance in difficult scenarios like sequential unlearning and against relearning attacks.
*   **Superior Data Efficiency:** Performing strongly even in low-resource settings and uniquely enabling effective **zero-shot unlearning** through interpretable SAE features.
*   **Interpretability:** Offering a more transparent unlearning mechanism by targeting specific, semantically meaningful SAE features.

were highlighted as contributions. We believe these characteristics make DSG a valuable and practical addition to the toolkit for LLM safety, privacy, and maintenance.

Reviewers JnnW and teuD both raised questions about DSG's effectiveness when forget and retain sets have a greater degree of overlap. In our responses, we have clarified that such overlap is indeed present in the MUSE benchmark variants used in our experiments, where DSG maintains strong performance.

---

### Decision · Program_Chairs · 2025-07-08

**Decision:**

Accept

**Comment:**

The paper introduces Dynamic SAE Guardrails (DSG) method for efficient and robust machine unlearning in LLMs with added interpretability. DSG leverages Fisher Information for feature selection and dynamic activation clamping within SAE. The experimental results surpass gradient-based unlearning on various aspects, making DSG as a practical and scalable progress with theoretical justification.


All reviewers point out that DSG outperforms gradient-based unlearning showing stronger unlearning with less utility loss on WMDP and MUSE benchmark. Most reviewers agree that DSG has theoretical basis grounded on Fisher Information and more interpretability from sparse autoencoders with dynamic clamping. Extensive ablation studies, additional sequential unlearning tests, and resistance to relearning attacks provide additional insights.


Many reviewers concern that all results are based on Gemma-2-2B. For thoroughness, testing larger models and diverse architectures are crucial. Some reviewers doubt whether robust and interpretable SAEs are always accessible. Two reviewers ask DSG's guardrails could be bypassed or degraded in white-box scenarios.


As many open LLMs are based on Llama family, scaling DSG to Llama's 7B models or state-of-the-art proprietary models is urgent, which can erase most concerns given the performance boost (especially on MUSE as a reviewer pointed out). Another key contribution is to set new research direction for activation-based, interpretable, and scalable safety interventions especially with respect to domain/task characteristics. LLM benchmarks (like BiGGenBench) classifies core capabilities and tasks for general LLMs, possibly inspiring more fine-grained experiments of unlearning across different domains/tasks. Other weaknesses could be addressable in future revisions provided with these directions. Note also that your draft has a space to include a few more relevant work pointed out by several reviewers.